# The 2018 European heatwave led to stem dehydration but not to consistent growth reductions in forests

Heatwaves exert disproportionately strong and sometimes irreversible impacts on forest ecosystems. These impacts remain poorly understood at the tree and species level and across large spatial scales. Here, we investigate the effects of the record-breaking 2018 European heatwave on tree growth and tree water status using a collection of high-temporal resolution dendrometer data from 21 species across 53 sites. Relative to the two preceding years, annual stem growth was not consistently reduced by the 2018 heatwave but stems experienced twice the temporary shrinkage due to depletion of water reserves. Conifer species were less capable of rehydrating overnight than broadleaves across gradients of soil and atmospheric drought, suggesting less resilience toward transient stress. In particular, Norway spruce and Scots pine experienced extensive stem dehydration. Our high-resolution dendrometer network was suitable to disentangle the effects of a severe heatwave on tree growth and desiccation at large-spatial scales in situ, and provided insights on which species may be more vulnerable to climate extremes.

---

Extreme climatic events are expected to become more common in a warming world[1]. Frequency and intensity of heatwaves, broadly defined as periods of consecutive days with anomalously high temperatures, have increased during the last decades and are expected to continue increasing throughout the 21st century[2]. Severity of heat extremes has particularly increased across European regions[3]. For instance, the length of summer heatwaves has doubled and the number of days registering heat extremes has tripled in western Europe since 1880[4]. Heatwaves rarely occur as pure temperature extremes but are often accompanied by anomalies in other climate parameters ("compound events")[5], such as a lack of precipitation and high evaporative demand. Drought stress thereby exacerbates the negative effects of extreme temperatures on tree productivity, vigour and survival[6]. The European heatwave in 2003, for instance, reduced ecosystem gross primary production over the continent by 30%[7]. Under combined drought and heat stress, stomatal closure and the associated inhibition of photosynthesis limit ecosystem carbon uptake. As soils dry out and canopy transpiration exceeds root water uptake, tree water reservoirs progressively deplete. Impoverishment of stem water status constrains growth[8,9], further reducing forests' potential to sequester carbon in woody biomass. In the short term, water release from internal stem reserves can temporarily buffer the negative effects of drought on the integrity of the tree's vascular system[8,10]. However, extended drought episodes will eventually cause hydraulic failure, together with tissue dehydration and damage, which may result in drought-induced tree mortality[11,12].

High-resolution dendrometers can capture complex signals integrating tree stem irreversible growth (GRO hereafter) and reversible radial fluctuations due to stem water release and refill. The latter mostly reflect bark tissue shrinking and swelling, which commonly follows a sub-daily pattern. Prolonged elastic reductions in stem diameter as drought proceeds are commonly referred to as tree water deficit[13] (TWD hereafter). Therefore, sub-daily measurements of stem diameter variations from dendrometers can provide valuable in situ metrics on the long-term physiological response of trees to changing climate in terms of growth and water status[8,14,15].

The summer of 2018 was exceptionally hot and dry in north-western Europe, whereas southern regions experienced relatively cooler and wetter conditions[16–18]. Large-scale assessments of forest productivity and sensitivity to environmental stresses during the 2018 heatwave (HW2018) have received great attention[18], with approaches ranging from multi-temporal satellite images[16,19–21], ecosystem-level carbon fluxes[20,22] and process-based model simulations[20,23]. Each of these approaches has its advantages and shortcomings, but none provides information on tree-level physiological responses to such increasingly frequent heatwave events. Large-scale analyses of high temporal-resolution dendrometer records could yield more mechanistic insight into drought impacts on tree growth and desiccation along environmental gradients. However, the lack of harmonised datasets has precluded composite analysis of regional-scale dendrometer data to date.

Here, we evaluate the effect of HW2018 on GRO and TWD across 21 widespread European tree species using a network of high-temporal resolution dendrometer and environmental data. Specifically, we hypothesised that:

(1) Relative to previous years, HW2018 will limit annual GRO and increase TWD, with the magnitude of these changes depending on site-specific environmental conditions.
(2) TWD as an index of drought stress will be lower in conifers compared to broadleaf species, as conifers commonly exhibit a relatively strong stomatal control and a conservative water-use strategy[24] to avoid increases in xylem tension that could result in hydraulic failure.

To test these hypotheses, we compiled high-resolution dendrometer records from 377 trees that met our initial requirements for data quality and temporal coverage (see Methods). Data were collected from 53 sites in mostly Central and Atlantic Europe (Fig. 1), i.e., in areas where the HW2018 was particularly intense[16,18,25]. During the heatwave timeframe (from day of year (DOY) 208 until 264; Supplementary Fig. 1), only three sites located in Romania experienced lower atmospheric and soil drought in 2018 compared to 2017 (Supplementary Fig. 2), another year of remarkable hot droughts in southern Europe[26]. A total of 21 broadleaf and conifer species were monitored during three consecutive years (2016–2018), with *Fagus sylvatica*, *Quercus spp.* (including *Q. petraea* and *Q. robur*), *Picea abies* and *Pinus sylvestris* being best represented (Supplementary Table 1).

From the individual dendrometer time series, we derived tree-specific daily cumulative GRO and daily extremes of TWD (minimum and maximum TWD, see Fig. 1c, d) during the HW2018 timeframe. Minimum TWD, commonly measured at night-time, was considered a baseline for comparison, as it is buffered against diurnal variability due to day-time transpirational water loss and thus reflects seasonal drought stress, which is largely controlled by available soil moisture. Maximum TWD was considered to additionally incorporate sub-daily stem shrinkage related to transpiration and diurnal drought stress dynamics, which are more directly linked to short-term fluctuations of atmospheric water demand and hence temperature. Absolute values of TWD during HW2018 and annual GRO varied substantially across and within species (Supplementary Fig. 3). Such variability, of one order of magnitude, called for the use of tree-specific ratios to evaluate the effect of HW2018 on TWD and GRO. Tree-specific normalisation of 2018 data relative to control years (see Methods) accounted for variability related to individual stem size, bark thickness, local environmental conditions, and wood traits that might affect depletion and refilling rates of stem water reserves. For instance, these include xylem hydraulic conductivity, hydraulic capacitance, wood elasticity, and xylem resistance to embolism formation. Tree-specific response ratios therefore allow for comparison of TWD and GRO across species and sites, highlighting differential temporal dynamics due to the HW2018. Unexpectedly, consistent reductions in stem growth were not observed across species and sites during 2018 but tree stems experienced greater shrinkage, with different sub-daily patterns found between conifer and broadleaf species.

## Results

Minimum TWD averaged across the heatwave timeframe in 2018 relative to the two preceding control years almost doubled (back-transformed minimum $TWD_{2018:control} = 1.8$ SE = 1.1, $P < 0.001$, Fig. 2a, Supplementary Table 2), with no differences between broadleaves and conifers ($P = 0.4$). Likewise, maximum TWD in 2018 was greater than during the control years (maximum $TWD_{2018:control} = 1.6$ SE = 1.1, $P < 0.001$), and similar for both taxonomic clades ($P = 0.5$). By contrast, the average annual growth did not consistently differ between 2018 and the control years ($GRO_{2018:control} = 0.9$ SE = 1.1, $P = 0.1$, Fig. 2b, Supplementary Table 2), nor was $GRO_{2018:control}$ different for broadleaf and conifer species ($P = 0.1$). When selecting sites that experienced on average higher vapour pressure deficit (VPD, as a proxy for atmospheric drought) and lower relative extractable water (REW, as a proxy for soil drought) during 2018 compared to the previous year (Supplementary Fig. 2), a non-significant trend of lower GRO in 2018 than during control years was detected

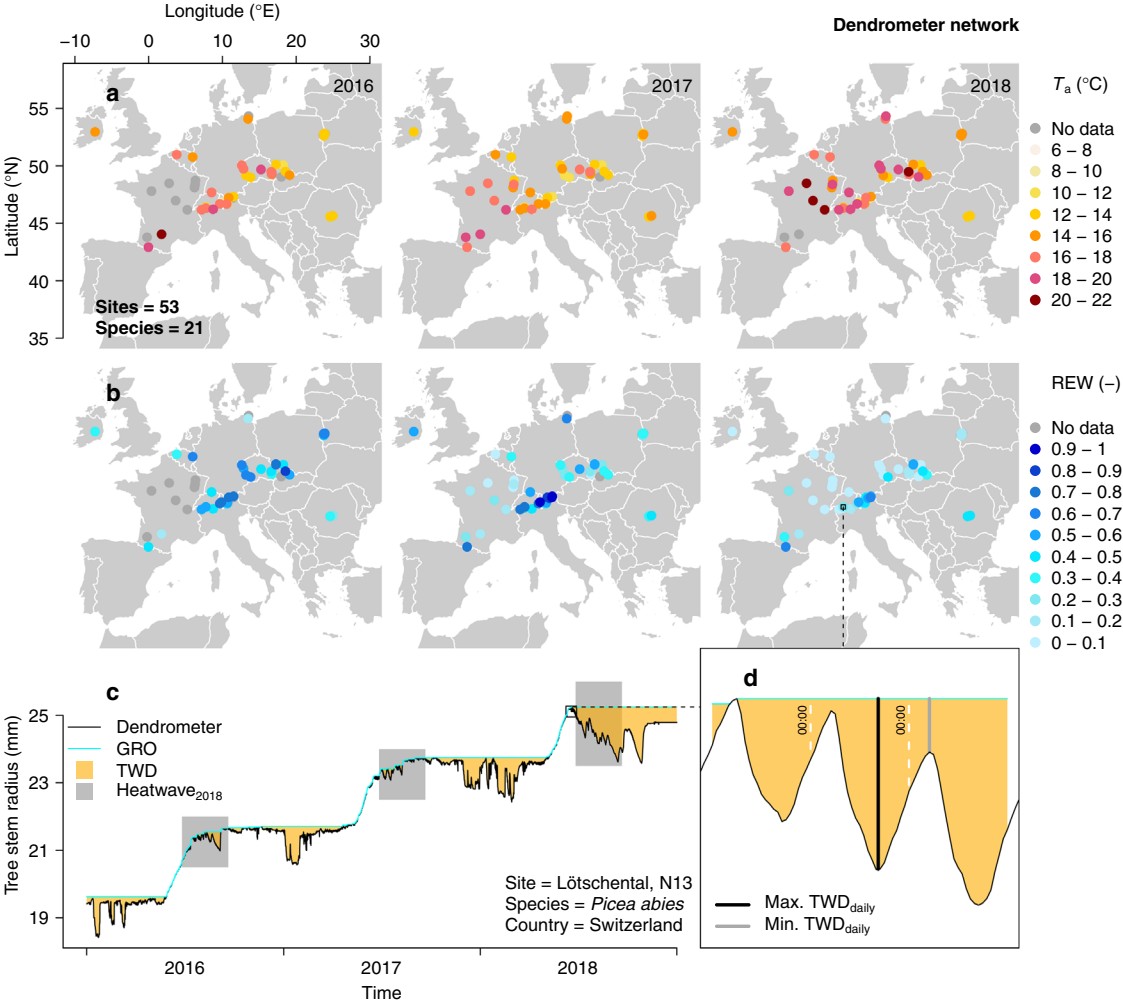

**Fig. 1 Climatic conditions during the 2018 heatwave period and dendrometer network. a**, **b** Spatial distribution of dendrometer sites and their respective mean atmospheric temperature ($T_a$ in °C) and relative extractable water (REW) during the 2018 heatwave timeframe (day of year 208 until 264) for 2016–2018. Sites with incomplete dendrometer time series data are indicated with grey dots. **c** Tree stem radius monitored at the Swiss Lötschental (site N13) for a *Picea abies* (L.) Karst. tree. The period corresponding to the 2018 heatwave is shown in all three years (defined as Heatwave$_{2018}$), in addition to tree water deficit (TWD) and extracted growth (GRO). **d** Inset of three days of tree stem radius monitored for the *P. abies* tree, where the concept of daily minimum and maximum TWD is shown (Max. TWD$_{daily}$ and Min. TWD$_{daily}$, respectively).

($P = 0.06$). Response metrics minimum TWD$_{2018:control}$ and GRO$_{2018:control}$ were inversely related across species ($P = 0.03$).

To isolate the tree-specific response from the site-specific environmental stress and compare between broadleaf and conifer species, daily minimum and maximum TWD$_{2018:control}$ were regressed against absolute daily VPD and REW during HW2018, hereafter referred to as hydrometeorological space. This procedure was used to ensure the comparison of daily minimum and maximum TWD$_{2018:control}$ under comparable climatic conditions and is presented in a 3-dimensional space using linear mixed effect models with a polynomial structure (see Methods). Both daily minimum and maximum TWD$_{2018:control}$ showed significant responses to decreasing REW and increasing VPD for broadleaf and conifer species ($P < 0.05$; Supplementary Table 3), with greater TWD$_{2018:control}$ ratios during periods of lower soil water availability and higher atmospheric evaporative demand (Fig. 3a–d). The hydrometeorological space in which daily minimum TWD was larger during 2018 compared to the 95$^{th}$ percentile of the control period (daily minimum TWD$_{2018:control}$ > 1) was approximate twice the size for conifers (23%) relative to broadleaves (10%; Fig. 3e). Contrastingly, daily maximum TWD$_{2018:control}$ across the hydrometeorological space was similar

between broadleaves and conifers (Fig. 3e). Therefore, differences between minimum and maximum TWD$_{2018:control}$ (sub-daily TWD amplitude hereafter), which denote sub-daily dynamics of stem day-time shrinkage and night-time swelling (Fig. 1d), were higher in broadleaves compared with conifers across the hydrometeorological space. This means that broadleaves had a greater capacity to refill stem water reservoirs during night-time, when the atmospheric evaporative demand was low. Note that the hydrometeorological space was similar for broadleaf and conifer species (Supplementary Figs. 4 and 5). Stem shrinkage patterns were further assessed in relation to HW2018 relative intensity. For this, VPD$_{2018:control}$ and REW$_{2018:control}$ were estimated considering a longer control period (2000–2017) to better capture background climate conditions. Consistent shrinkage patterns were found (Supplementary Fig. 6), although the overlap in the common hydrometeorological space was limited, and differences in sub-daily TWD amplitude between broadleaves and conifers were reduced.

The response of daily minimum TWD$_{2018:control}$ to absolute daily REW and VPD was additionally isolated for the four most abundant (Supplementary Table 1) and commercially relevant European tree species[27] in the dendrometer network (Fig. 4). The

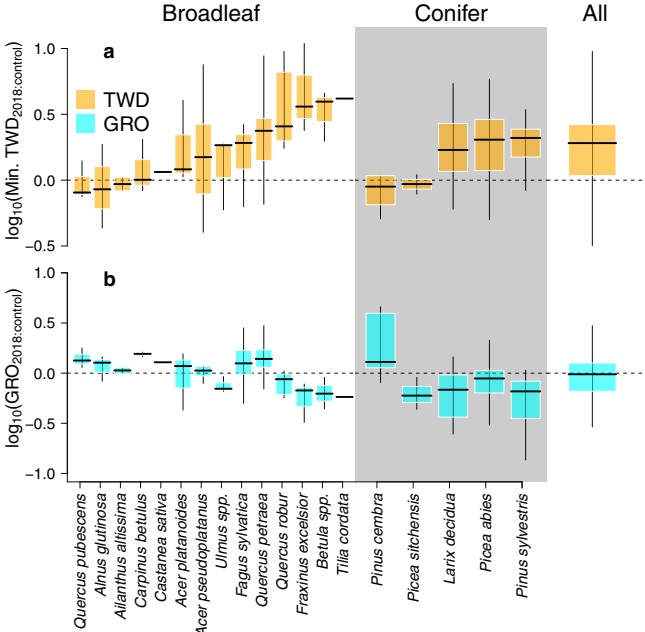

**Fig. 2 Species-specific ratios of tree water deficit (TWD) during the 2018 heatwave and 2018 annual growth (GRO) relative to control years (2016–2017). a, b** Boxplots of broadleaf and conifer minimum tree water deficit (min. $TWD_{2018:control}$; **a**) and annual radial stem growth ($GRO_{2018:control}$; **b**) in response to the 2018 heatwave relative to control years. Log transformed ratios are shown to linearise and normalise the response metric. $\log_{10}(TWD_{2018:control})$ above zero indicates a larger shrinkage was registered during the 2018 heatwave compared to the control years, whereas $\log_{10}(GRO_{2018:control})$ near zero indicates that stem growth in 2018 was similar to that in control years. Centerlines, box limits, and whiskers represent the median, upper and lower quartiles, and extremes excluding outliers (those further than the 1.5x interquartile range). $n = 175$ tree stems over 37 sites.

highest values of minimum $TWD_{2018:control}$ were found for oak trees (> 2, Fig. 4b). However, such severe shrinkage was uniquely observed in a small area of largely dried-out soils (REW < 0.1), regardless of VPD (Supplementary Table 4). Most of the oak hydrometeorological space showed the lowest values of minimum $TWD_{2018:control}$ (74% with values < 0.5, Fig. 4e), suggesting a strong capacity of this species to maintain stem hydration during HW2018 across the monitoring network. On the contrary, both conifer species (Norway spruce and Scots pine) experienced moderate to high minimum $TWD_{2018:control}$ across most of their hydrometeorological space (74–77% with values > 0.5, Fig. 4e), with greater relative stem shrinkage with increasing atmospheric and soil drought (Fig. 4c–d, Supplementary Table 4). European beech exhibited an intermediate behaviour between oak and conifer species (Fig. 4a).

## Discussion

### Overall stem growth and dehydration responses to the heatwave.
Our results partially supported hypothesis 1, as stems experienced larger shrinkage during HW2018 relative to control years. However, no consistent reductions in annual stem growth were found (Fig. 2), as a variable response among sites and species canceled each other out. Illustratively, regional studies have reported negative[28,29], neutral[30,31], and positive[32] responses of annual stem growth to HW2018. Similarly, our results were dependent on the site selection for analysis, as more restrictive criteria according to HW2018 intensity tended to increase the plausibility of growth reductions. Limitations to stem-girth

increment are therefore highly dependent on site-specific conditions and, importantly, on the timing of the extreme climatic event. Early phenological phases of wood formation, namely cell division, and enlargement, result in stem-girth increment. Later during the growing season, the formation, filling, and lignification of secondary cell walls increase the density of the newly forming biomass but do not translate into detectable volumetric growth[33]. The HW2018 started in late July at most of the monitored sites (Supplementary Fig. 1), when cell division and enlargement phases were probably about to cease. Hence, the vast majority of the annual stem-girth increment was already formed at this time, which may explain the absence of a clear heatwave effect on 2018 stem volumetric growth. Most likely, only drought stress during spring and early summer can effectively limit current-year volumetric growth[34,35], while drought later in the season might reduce wood density to a greater extent[36]. Our tree-level observations agree with ecosystem-level measurements of carbon exchange. Ecosystem carbon uptake across central and northern Europe was stimulated during spring 2018, which was characterized by relatively warm and humid conditions[17,18,20,23]. The transition into an extreme summer drought (HW2018) rapidly reduced the strength of ecosystem carbon sink[17,18,20,23], likely due to hydromechanical restrictions to cell wall deposition and lignification[8,36]. At the annual scale, spring growth stimulation and summer reduction compensated each other, thereby resulting in nearly average ecosystem carbon uptake over the year[23].

In contrast to GRO, greater stem shrinkage in 2018 relative to the control years was detected in both broadleaf and conifer species, indicating increased stem dehydration. Stimulation of canopy leaf area due to favorable spring conditions for foliage development, leading to enhanced transpiration, likely exacerbated drought effects of HW2018 on soil and stem water status[17]. Stem dehydration during HW2018 progressively reduced the trees' ability to release stored water into the transpiration stream, likely leading to substantial losses in hydraulic conductivity[8,10]. Accordingly, leaf cellular damage and premature leaf senescence as initial symptoms of partial dysfunction of the hydraulic system[6] were detected at larger-spatial scales. Remote-sensing observations in central and northern Europe have reported anomalous reductions in normalised difference vegetation index[16,21,25] and leaf area index[19] during summer 2018. Such a reduction in summer vegetation greenness reflects an early leaf fall and even partial or complete canopy dieback[21,25]. Moreover, stem dehydration during HW2018 may have slightly shortened phases of cell division and enlargement[35], as suggested here by the inverse relationship between $TWD_{2018:control}$ and $GRO_{2018:control}$. In the medium and long term, legacy effects of HW2018 are expected to further dampen stem growth[37,38] and potentially trigger tree decline, as recently observed in some areas of central Europe[25].

### Stem dehydration across the hydrometeorological space.
Contrary to our second hypothesis, a more conservative water-use strategy of conifer species[24] did not confer a greater capacity to maintain stem water reserves during HW2018 (Fig. 3), as similarly found before among pine, spruce, and oak trees[39]. Conifers showed larger daily minimum $TWD_{2018:control}$ than broadleaf species under comparable hydrometeorological conditions, denoting greater stem dehydration during the heatwave relative to the control period. More importantly, the sub-daily TWD amplitude was relatively low for conifers across the hydrometeorological space, indicating limited refilling of internal stem water reserves on a sub-daily basis. These results could be explained by the low xylem-specific hydraulic conductivity observed in conifer woods[40], a lower leaf minimum conductance

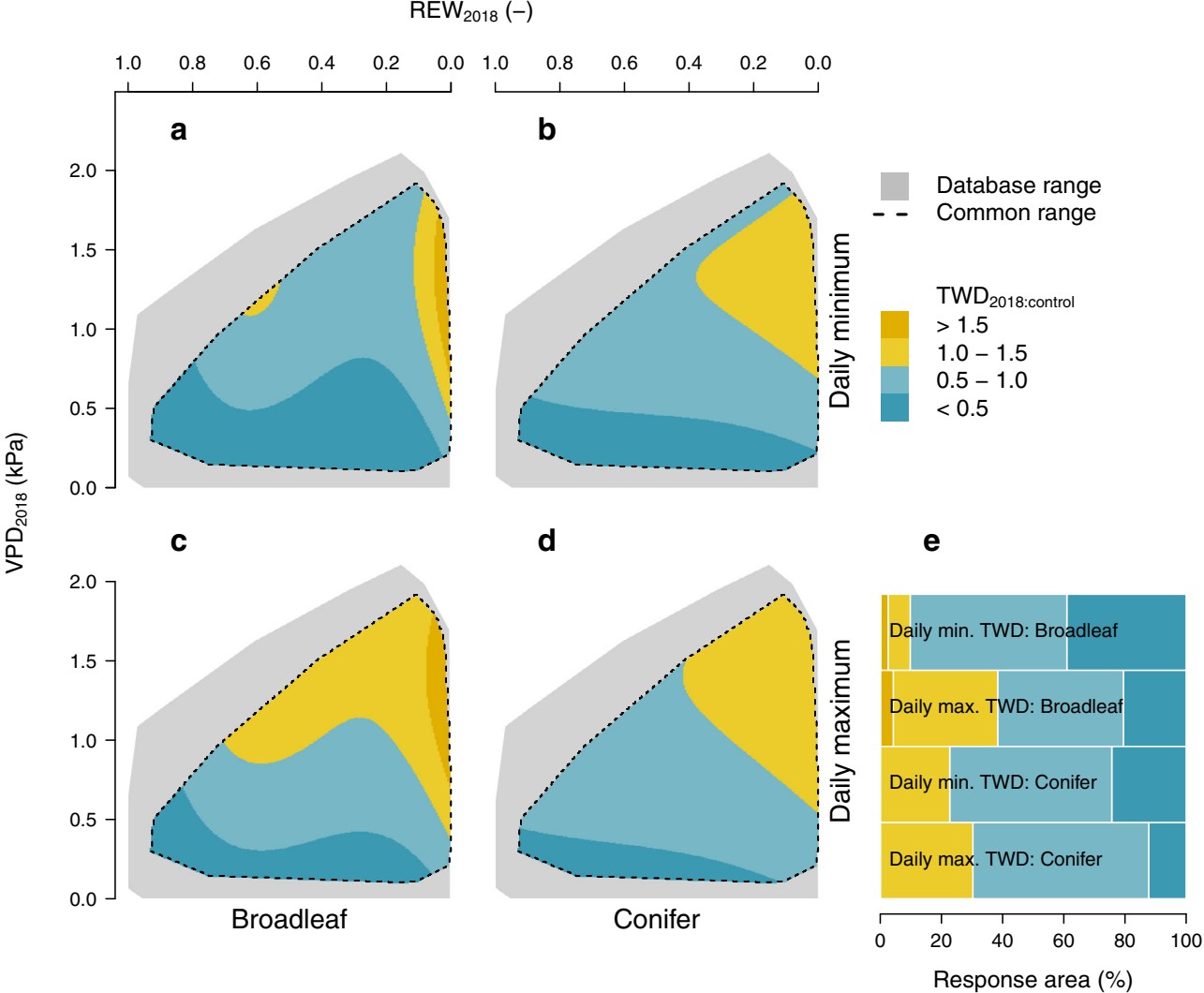

**Fig. 3 Response of tree water deficit (TWD) ratio to vapor pressure deficit (VPD, kPa) and relative extractable water (REW, unitless [-]) of broadleaf (a, c) and conifer (b, d) species in the hydrometeorological space. a–d** Linear-mixed effect model output of the ratio of the daily minimum (**a, b**) and maximum (**c, d**) TWD during the 2018 heatwave compared to the 95[th] percentile of the control period (TWD$_{2018:control}$), while VPD$_{2018}$ and REW$_{2018}$ refer to the absolute values during the 2018 heatwave. TWD$_{2018:control}$ above 1 indicates that a larger shrinkage was registered during the 2018 heatwave relative to the control period. From the hydrometeorological space range of the entire database (indicated in grey), models have been adjusted for the common climatic range of broadleaf and conifer species only (indicated with dotted lines). **e** Bars indicate the percentage of the hydrometeorological space covering different ranges of TWD$_{2018:control}$ values (see legend).

under high VPD[41], or a hydraulic disconnection from the atmosphere and the rizhospere[42,43]. This strategy comes however at the expense of limited refilling of stem water reserves through root water uptake during night-time, as previously observed for Scots pine trees[42]. Note that compiled data from conifer species is limited here to the Pinaceae family (Fig. 2), whose hydraulic behaviour diverges markedly from that of other families (e.g., Cupressaceae)[44]. By contrast, broadleaves tend to close stomata closer to critical thresholds for hydraulic functionality[43,45], allowing for continued water use and carbon gain under prolonged periods of atmospheric and soil drought. Thus, broadleaves disconnect less easily from drying soils and demanding atmospheres[43], which leads to the development of water potential gradients throughout the plant to replenish stem water reserves at night when root water uptake exceeds transpiration loss. Furthermore, maintenance of the soil-plant-atmosphere hydraulic continuum for transpirational cooling of the leaves under heat stress[46] might be more important for broadleaves, characterized by thin leaves with a large area per unit of mass exposed to

thermal stress. This relatively risky hydraulic strategy of deciduous broadleaves might be related to their ability to shed leaves to avoid further dehydration under extreme drought stress, as foliage and hydraulic functionality can be restored during the next vegetative season. Contrastingly, most conifer species cannot afford such penalty in terms of carbon loss, as the high investment in needle development compels evergreen trees to maintain their foliage over several years[37]. Therefore, night-time replenishment of internal stem water reserves in broadleaves, reflected here by a relatively large sub-daily TWD amplitude, denotes partial and transient recovery and relaxation from environmental stress. On the other hand, prolonged exposure to tissue dehydration and damage during seasonal drought in conifers might contribute to relatively longer and stronger drought legacy effects of up to several years observed across forest biomes[38,47].

Inter-specific comparisons showed that oak stems remained relatively hydrated across the hydrometeorological space during HW2018 (Fig. 4). Similarly, oak trees showed remarkable drought resistance and were able to maintain constant transpiration

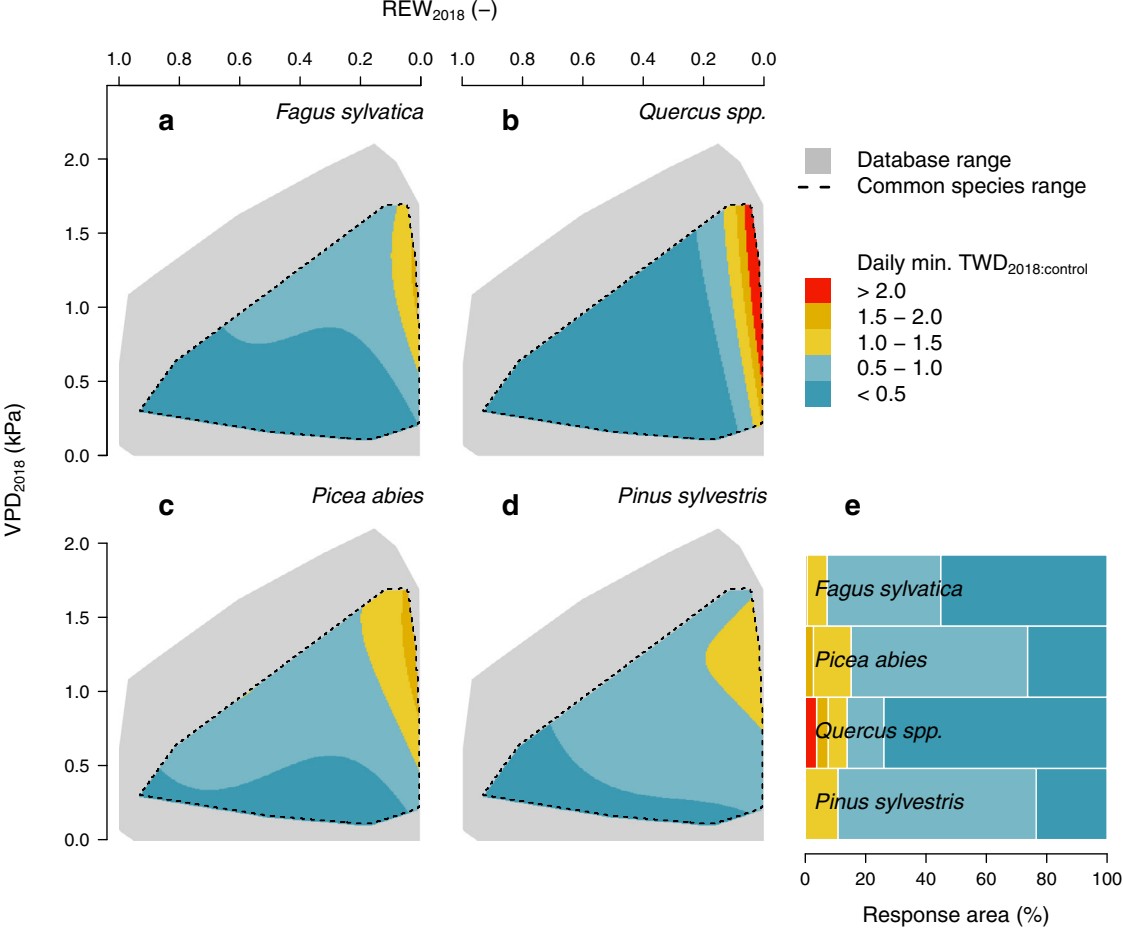

**Fig. 4 Species-specific response of tree water deficit (TWD) ratio to vapour pressure deficit (VPD) and relative extractable water (REW) in the hydrometeorological space for most abundant and economically valuable tree species in Europe. a–d,** Linear-mixed effect model output of the ratio of the daily minimum TWD during the 2018 heatwave compared to the 95$^{th}$ percentile of the control period (TWD$_{2018:control}$), while VPD$_{2018}$ and REW$_{2018}$ refer to the absolute values during 2018 heatwave. All trees of *Fagus sylvatica*, *Quercus spp.* (*Q. robur/petraea*), *Picea abies* and *Pinus sylvestris* within the database are considered. From the hydrometeorological space range of the entire database (indicated in grey), models have been adjusted for the common climatic range of the selected species only (indicated with dotted lines; see Supplementary Fig. 4 for the climate range of each species). **e** Bars indicate the percentage of the hydrometeorological space covering different ranges of min. TWD$_{2018:control}$ values (see legend).

throughout the 2003 European heatwave, e.g., in a Swiss mixed forest[48]. Root access to deep soil water might be a crucial factor driving the species-specific response to water shortage[49], so that deep-rooted species (oak) might be able to maintain a better water status than shallow-rooted ones (e.g., Norway spruce)[50]. Variability in the daily minimum TWD$_{2018:control}$ across the hydrometeorological space might also depend on species-specific plasticity of leaf and stomatal physiology to optimise tree water use according to site conditions. In addition, the contribution of stem water release to tree transpiration is not necessarily uniform across species and environmental gradients[10,51], and tree transpiration could be even decoupled from TWD, as already observed among four temperate species[49]. Other hydraulic traits are likely involved in the complex species-specific response of stem water reserves to heatwaves. For instance, we could expect that tree species with higher stem hydraulic capacitance and less dense wood, and hence a lower Young´s modulus of elasticity, would experience larger stem shrinkage and swelling for a given change in xylem tension. Linking our dendrometer findings to leaf[52], root[53] and plant-specific hydraulic traits[54] would be a critical next step, yet requires data from more species with significant trait variability and greater spatial detail. The limited ability of Norway spruce and Scots pine to maintain minimum levels of stem hydration during periods of severe drought stress

(Fig. 4) are consistent with relatively strong drought legacy effects among *Pinaceae* species[38] and, at larger spatio-temporal scales, with projected shifts in species distribution across Europe[27,55]. Although there is still substantial uncertainty regarding how inter-specific differences in water economy scale in terms of productivity and tree mortality risk, the capacity to maintain minimum levels of tissue hydration for extended dry periods seems to be crucial to determine how different species respond to adverse environments and extreme events in the long term[11,12,56]. Biophysical responses related to tree water relations likely precede any structural adjustment to drought stress, as observed in pine trees after stopping 11 years lasting irrigation treatment, where growth decline lagged behind increases in TWD by several years[37]. Similarly, stem shrinkage and near-zero sub-daily TWD amplitude preceded hydraulic dysfunction and partial or complete canopy dieback in pine and beech trees[57,58]. We argue that stem diameter variations might be employed as an early-warning signal for long-term reductions in tree productivity evidencing tree decline.

**Outlook**. Large-scale analyses of dendrometer data are challenging. Sensor maintenance requires frequent fieldwork, auto-matised routines for data processing have just recently become

available (see Methods), and harmonisation of highly-resolved and long-term records from different dendrometer types further hinders the compilation of homogeneous global datasets. We believe, however, that dendrometer networks will strengthen as the unparalleled potential of stem diameter variations to capture in situ forest productivity and sensitivity to environmental stresses is broadly recognised. Compilation of large-scale datasets of dendrometer data, together with those of tree transpiration[59], open promising research avenues, as tree-level datasets can be spatially coupled with monitoring networks of ecosystem carbon and water fluxes[60] and remote-sensing observations. Integration of tree-, regional-, and global-based data sources will advance knowledge on the mechanisms underlying tree response to climate change and extremes.

Our analyses draw attention to the potential of dendrometer data for use as an early warning system to detect stress thresholds for tree vitality and growth at large spatial scales in situ, and hence identify areas with a high risk of forest decline and mortality. We call for a tree-centered approach, with stems as the main source of information on species- and site-specific responses to different stress ellicitors[8,60,61]. Here, highly-resolved dendrometer time series, mostly covering Central and Atlantic Europe, have shown the overall limited effect of the HW2018 on current-year volumetric growth, despite the widespread depletion of stem water reserves during the heatwave period. Carbon investment for canopy development in spring 2018 may have yielded a poor photosynthetic return during summer, when trees operated close to their dehydration thresholds, resulting in a likely reduction in tree carbon reserves over the year. Long-term legacy effects due to the depletion of carbohydrate reserves and damage to the hydraulic system[37,38] during HW2018 will very likely compromise tree growth, performance, and survival in the coming years[62]. Furthermore, contrasting stem water refilling behaviour between broadleaves and conifers links to differences in stomatal regulation[24] and hydraulic safety margins[43,45] observed between taxonomic clades and broadens our perspective on tree hydraulic functioning.

## Methods

Tree-specific point and band dendrometer measurements with a temporal resolution of 15–60 min were compiled from 85 monitoring plots across Europe. Plots within a Euclidean distance of 9 km, an elevational difference of less than 300 m, and similar soil water conditions were clustered, resulting in a total of 53 sites (Supplementary Table 1). For each monitored tree, information on the (i) species, (ii) site location (coordinates in °E and °N), (iii) radius time-series (in μm), (iv) timestamp with recorded time-zone, (v) quality assessment, and (vi) climate was collected. A quality assessment of the radius time series was performed to assist analyses with specific data-quality requirements (see details below), flagging trees with (a) plateauing values during periods longer than seven days (a common issue of band dendrometer data), and (b) temporal gaps larger than 14 days.

Site-specific meteorological data were compiled to determine the HW2018 timeframe and perform climate-response analyses. For each site, we extracted time series of daily mean atmospheric temperature ($T_a$ in °C) and vapour pressure deficit (VPD in kPa), obtained from the nearest climate station (search radius = 80 km) using the Global Surface Summary of the Day (GSOD) Weather Data Client[63]. Gaps were filled by linear interpolation with in situ measurements (if available) when the daily time series showed a high correlation (Pearson's rho > 0.7). Site-specific soil moisture data were obtained from the ERA-5 land surface model simulations[64] (spatial resolution = 9 km; temporal coverage = 2015–2019; Layer 3: 28–100 cm depth). Relative extractable water (REW) was calculated to account for inter-site differences in soil properties and absolute soil moisture levels and facilitate inter-site comparisons[65]. REW was calculated by scaling the site-specific daily soil moisture simulations to the field capacity and the site lowest soil moisture value according to:

$$REW_{j,i} = \frac{SM_{j,i} - SM_{min,j}}{SM_{95th,j} - SM_{min,j}}$$

where $REW_{j,i}$ and $SM_{j,i}$ are site (j) and daily (i) values, and $SM_{95th,j}$ and $SM_{min,j}$ are the site-specific soil moisture approaching field capacity and minimum values, respectively. The 95th percentile, and not the maximum value, was applied to exclude heavy rain events that may result in soil water saturation (e.g., Supplementary Fig. 7). Long-term

climatological conditions, including mean annual temperature and mean annual precipitation (Supplementary Table 1), were obtained from CHELSA[66] (spatial resolution 1 km). To establish the heatwave timeframe, longer daily time series of the maximum daily air temperature were obtained from E-OBS[67] (spatial resolution = 0.1°, temporal coverage = 1951–2018). Heatwave days were defined as five consecutive days with a maximum daily temperature higher than the 90th percentile of the control period (1951–2000)[68,69]. The overall heatwave extent was established from DOY 208 until 264, during which period heatwave days overlapped for more than five sites (Supplementary Fig. 1).

The dendrometer time series were checked and homogenised using the treenetproc R package (version 0.1.4)[70]. The cleaned dendrometer time series were partitioned into growth- and water-related components of stem radius variation according to the zero-growth concept[13]. This procedure assumes that growth (GRO) starts once the previous stem diameter maximum is exceeded and ends as soon as the stem starts shrinking. Diameter variations below the preceding maximum stem diameter are considered as a period of tree water deficit (TWD), a proxy of stem dehydration and tree drought stress[14]. In short, TWD is a measure for water depletion, mainly in the living bark tissues of the stem, expressed as stem shrinkage (in μm), which occurs when canopy transpiration exceeds root water uptake, hence retrieving water from stem water reserves to meet the evaporative demand. For GRO, daily time series of cumulative annual growth was established. For TWD, daily minimum and maximum time series were established to capture night-time and day-time water status according to sub-daily fluctuations in the atmospheric evaporative demand. To isolate the temporal variability in GRO and TWD in 2018 relative to control years (2016–2017) and facilitate comparison among trees, we estimated tree-specific ratios of GRO and TWD in 2018 divided by control years. TWD ratios were estimated extracting daily TWD time series covering the HW2018 timeframe (DOYs 208-264) for 2018 and control years. Two approaches, with different requirements for the minimal extent of the control period (see below), were applied to analyse the dendrometer time series, considering annual and daily temporal scales.

The first approach evaluated the HW2018 impact on GRO and TWD on an annual basis (Fig. 2, Supplementary Table 2). Annual cumulative GRO and the daily minimum and maximum TWD averaged over the course of the HW2018 timeframe (DOYs 208-264) were estimated per year. Here, solely tree-specific time series with overall appropriate quality and covering 2016–2018 were considered for analyses. Only two adjacent years (2016 and 2017) were considered as control years to minimise time-related changes in tree structure and function, while maximising available data. Moreover, data before 2016 were excluded to avoid biases due to different lengths of the time series. Ratios of tree-specific GRO and minimum and maximum TWD between 2018 and control years were then calculated and log-transformed ($\log_{10}[GRO_{2018:control}]$, $\log_{10}[\min. TWD_{2018:control}]$ and $\log_{10}[\max. TWD_{2018:control}]$, respectively), which is a standard measure to quantify effect size in meta-analyses[71]. Before $\log_{10}$ transformation, the unit (1 μm) was added to GRO to maintain trees with null growth within analyses. To test differences between taxonomic clades (broadleaves and conifers) on log-transformed $GRO_{2018:control}$ and min. and max. $TWD_{2018:control}$, linear mixed effect models (lme4 R package, version 1.1-21[72]) were fitted considering species and site as crossed random effects. Backward stepwise selection was applied, so taxonomic clade was omitted from the model if not significant (P > 0.05). Correlation between log-transformed $GRO_{2018:control}$ and min. $TWD_{2018:control}$ was tested likewise. Significance P values were calculated using the R package lmertest (version 3.1-1). Back-transformed model estimates are reported in the text.

In contrast to the annual values used in the first approach, the second approach assessed the daily response of minimum and maximum TWD to HW2018 (Figs. 3–4, Tables S3–S4). This analysis was performed to compare the climatic response of broadleaves and conifers along gradients of comparable absolute VPD and REW values experienced during the 2018 heatwave ($VPD_{2018}$ and $REW_{2018}$, respectively). Here, the ratio between the 2018 daily TWD and the 95th percentile of the control years (2016–2017) within the heatwave timeframe was calculated. The 95th percentile was selected for daily analyses to compare stem shrinkage during HW2018 relative to the nearly maximum experienced during the control period. Note that mean values used for annual analyses are more dependent on days with zero TWD and hence more suitable for long-term comparison of both shrinkage intensity and duration. For the climatic response analysis, linear mixed effect models were fitted to predict min. and max. daily $TWD_{2018:control}$ as a function of daily $VPD_{2018}$ (using a 2nd order polynomial) and $REW_{2018}$ (using a 3rd order polynomial), with tree nested within site, and species as crossed random (intercept) effects. Before the log transformation, 1% of the max. $TWD_{2018:control}$ was added to maintain zero values within the analyses. Only species present in more than one site were considered here. Additionally, similar analyses were separately performed for the four most important forest species of Europe, both in terms of spatial distribution and economic importance, within the database: Fagus sylvatica, Quercus spp. (including Q. robur and Q. petraea), Picea abies, and Pinus sylvestris. Moreover, these species were selected due to their appropriate spatial coverage within the available dataset, both in terms of number of sites and registered gradients of climatic conditions. All species-specific data were considered in species-specific models, even when quality and temporal extent criteria were not entirely satisfied. Less restrictive criteria were adopted here to maximize the size of the available datasets, as initial models did not converge due to the low spatial coverage for individual species. The fitted models were back-transformed for

calculating the hydrometeorological space covering different levels of min. and max. $TWD_{2018:control}$. All analyses were performed in the R software (version 3.6[73]).

**Reporting summary**. Further information on research design is available in the Nature Research Reporting Summary linked to this article.

## Data availability
The dendrometer data and the site metadata used in this study are available in the Zenodo repository, under accession code https://doi.org/10.5281/zenodo.5711706.

Site-specific meteorological data were compiled using the Global Surface Summary of the Day (GSOD) Weather Data Client (https://joss.theoj.org/papers/10.21105/joss.00177).

Site-specific soil moisture data were obtained from the ERA-5 land surface model simulations (https://www.ecmwf.int/en/era5-land).

Long-term climatological conditions were obtained from CHELSA at a 1 km spatial resolution 1 km (https://chelsa-climate.org/).

## Code availability
The codes generated for data analyses for the current study (R scripts) are available from the corresponding author on reasonable request.

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

## Acknowledgements
This work utilised the network of dendrometer observations established by the COST Action network STReESS (grant FP1106). We acknowledge the involved networks TreeNet, Swiss Long-term Forest Ecosystem Research Programme LWF, French National Network for Long-term FOrest ECOsystem Monitoring RENECOFOR, the German Long Term Ecosystem Research Network LTER-D, the Italian Long Term Ecosystem Research Network ILTER, the Integrated Carbon Observation System ICOS and Tree-Watch.net. R.L.S. acknowledges funding from the Special Research Fund (BOF) of Ghent University for Postdoctoral Fellowships and the Spanish Ministry of Science, Innovation, and Universities (Juan de la Cierva Programme, grant IJC2018-036123-I). R.L.P., R.Z., P.F., and G.v.A. acknowledge funding from the Federal Office for the Environment FOEN (00.0365.PZ I 0427-0562, 09.0064.PJ/R301-0223; project treenet.info) and the Swiss National Science Foundation SNF (20FI21_148992, 20FI_173691, P2BSP3_184475; project LOTFOR 150205 and Grant 20FI20_173691; project ICOS-CH). J.M.V. and M.M. acknowledge funding from the Spanish MINECO via competitive grants CGL2013-46808-R and CGL2017-89149-C2-1-R. R.P. acknowledges funding from the grant RTI2018-095297-J-I00 (Spain) and by a Humboldt Research Fellowship for Experienced Researchers (Germany). F.B acknowledges funding from the HOMING project "Inside out" (#POIR.04.04.00-00-5F85/18-00) by the Foundation for Polish Science under the European Regional Development Fund. G.J.N. acknowledges funding from the forest-climate pilots of the Netherlands funded by Ministry of Agriculture, Nature Management and Food Quality, the H2020 project VERIFY (776810). M.L. and A.I.S. acknowledge funding from the German Federal Ministry of Food and Agriculture (BMEL; project SURE; Forst-2017-2). E.C. acknowledges funding from SustES, Adaptation strategies for sustainable ecosystem services and food security under adverse environmental conditions (CZ.02.1.01/0.0/0.0/16_019/0000797). S.M. acknowledges funding from the "Austrian Science Fund" (FWF, project P29896-B22). N.D. acknowledges funding from the SACROBOQUE project (GIP-ECOFOR, grant #2016.013) and the FOREPRO project (ANR-19-CE32-0008). J.K. acknowledges funding from the Ministry of Education, Youth and Sports of CR (CzeCOS program, grant number LM2018123). W.O. acknowledges funding from the Austrian Science Fund (FWF, projects P22280-B16, P25643-B16). C.B.K.R acknowledges funding from the French National Research Agency (ANR) as part of the "Investissements d'Avenir" program (ANR-11-LABX-0002-01, Lab of Excellence ARBRE). B.T., B.O and M.S acknowledge funding from 'HEA, PRTLI (Cycle 5) infrastructure award'. J.U. acknowledges funding from Czech Science Foundation, project 21–11487 S. M.V. acknowledges funding by the Ministry of agriculture of the Czech Republic, institutional support MZE-RO0118. L.K and K.S. acknowledge funding from the Project LIFE + ForBioSensing PL (LIFE13 ENV/PL/000048) and Poland's National Fund for Environmental Protection and Water Management (contract 485/2014/WN10/OP-NM-LF/D). N.O. acknowledges funding from the LTSER platform LTER_EU_IT_097—Val Mazia/Matschertal (Province of Bozen/Bolzano—South Tyrol). J.B. acknowledges funding from the LOEWE priority program Nature 4.0 (UM2; Hessian State Ministry for Higher Education, Research and the Arts, Germany). M.S. acknowledges funding from the Ministry of Education, Youth and Sports of CR within the CzeCOS program (grant number LM2018123). K.S. and R.L.S acknowledge the contribution of Jonas von der Crone. N.D. acknowledges maintenance of micro-dendrometers by Jean-Marc Louvet, Sophie Lorentz, and French National Forestry Office (ONF) technicians.

## Author contributions
The initial concept of DenDrought2018 was conceived by USK, PF, KS, and ML. AIS and MS built the database, with the contribution of GvA, RP, FB, RLP, and RZ. RLS and RLP performed the analyses and drafted the first and final version of the paper, in close collaboration with RZ, USK, and KS AIS, MS, RP, FB, EC, PF, BL, ML, JMV, MM, GJN, EvM, and GvA contributed to the analyses and the preparation of the paper. The rest of the coauthors contributed data to the dataset and read and commented on the paper.

## Competing interests
The authors declare no competing interests.

## Additional information

Roberto L. Salomón [1,2,60], Richard L. Peters [1,3,60], Roman Zweifel [3], Ute G. W. Sass-Klaassen [4✉],
Annemiek I. Stegehuis [5,6], Marko Smiljanic [7], Rafael Poyatos [8,9], Flurin Babst [10,11], Emil Cienciala [12,13],
Patrick Fonti [3], Bas J. W. Lerink [14], Marcus Lindner [5], Jordi Martinez-Vilalta [8,9],
Maurizio Mencuccini [8,15], Gert-Jan Nabuurs [4,14], Ernst van der Maaten [16], Georg von Arx [3],
Andreas Bär [17], Linar Akhmetzyanov [4], Daniel Balanzategui [18,19], Michal Bellan [13,20], Jörg Bendix [21],
Daniel Berveiller [22], Miroslav Blaženec [23], Vojtěch Čada [6], Vinicio Carraro [24], Sébastien Cecchini [25],
Tommy Chan [26], Marco Conedera [27], Nicolas Delpierre [22], Sylvain Delzon [28], Ľubica Ditmarová [23],
Jiri Dolezal [29,30], Eric Dufrêne [22], Johannes Edvardsson [31], Stefan Ehekircher [32], Alicia Forner [33,34], Jan Frouz [35],
Andrea Ganthaler [17], Vladimír Gryc [36], Aylin Güney [37,38], Ingo Heinrich [18,19,39], Rainer Hentschel [40],
Pavel Janda [6], Marek Ježík [23], Hans-Peter Kahle [41], Simon Knüsel [27], Jan Krejza [13,20],
Łukasz Kuberski [42], Jiří Kučera [43], François Lebourgeois [44], Martin Mikoláš [6], Radim Matula [6],
Stefan Mayr [44], Walter Oberhuber [44], Nikolaus Obojes [45], Bruce Osborne [46,47], Teemu Paljakka [26],
Roman Plichta [48], Inken Rabbel [49], Cyrille B. K. Rathgeber [3,44], Yann Salmon [26,50],
Matthew Saunders [51], Tobias Scharnweber [7], Zuzana Sitková [52], Dominik Florian Stangler [41],
Krzysztof Stereńczak [53], Marko Stojanović [13], Katarína Střelcová [54], Jan Světlík [13,20], Miroslav Svoboda [6],
Brian Tobin [47,55], Volodymyr Trotsiuk [3,6], Josef Urban [48,56], Fernando Valladares [34], Hanuš Vavrčík [36],
Monika Vejpustková [57], Lorenz Walthert [3], Martin Wilmking [7], Ewa Zin [42,58], Junliang Zou [59] &
Kathy Steppe [1✉]

[1]Laboratory of Plant Ecology, Department of Plants and Crops, Faculty of Bioscience Engineering, Ghent University, 9000 Ghent, Belgium. [2]Grupo de Investigación Sistemas Naturales e Historia Forestal, Universidad Politécnica de Madrid, 28040 Madrid, Spain. [3]Swiss Federal Institute for Forest Snow and Landscape Research WSL, 8903 Birmensdorf, Switzerland. [4]Forest Ecology and Forest Management, Wageningen University and Research, 6700 AA Wageningen, The Netherlands. [5]European Forest Institute, Resilience Programme, 53113 Bonn, Germany. [6]Department of Forest Ecology, Faculty of Forestry and Wood Sciences, Czech University of Life Sciences Prague, 165 00, Prague, Czech Republic. [7]DendroGreif, Institute for Botany and Landscape Ecology, University Greifswald, 17487 Greifswald, Germany. [8]CREAF, E08193 Bellaterra (Cerdanyola del Vallès), Catalonia, Spain. [9]Universitat Autònoma de Barcelona, E08193 Bellaterra (Cerdanyola del Vallès), Catalonia, Spain. [10]School of Natural Resources and the Environment, University of Arizona, Tucson, AZ 85721, USA. [11]Laboratory of Tree-Ring Research, University of Arizona, Tucson, AZ 85721, USA. [12]IFER—Institute of Forest Ecosystem Research, 254 01 Jilove u Prahy, Czech Republic. [13]Global Change Research Institute of the Czech Academy of Sciences, 603 00 Brno, Czech Republic. [14]Wageningen Environmental Research, Wageningen University and Research, 6700 AA Wageningen, The Netherlands. [15]ICREA, 08010 Barcelona, Spain. [16]Chair of Forest Growth and Woody Biomass Production, TU Dresden, 01737 Tharandt, Germany. [17]Department of Botany, University of Innsbruck, 6020 Innsbruck, Austria. [18]Climate Dynamics and Landscape Evolution, Helmholtz Centre Potsdam, GFZ German Research Centre for Geosciences, 14473 Potsdam, Germany. [19]Geography Department, Humboldt University, 12489 Berlin, Germany. [20]Department of Forest Ecology, Faculty of Forestry and Wood Technology, Mendel University in Brno, 613 00 Brno, Czech Republic. [21]Laboratory for Climatology and Remote Sensing (LCRS), Faculty of Geography, 35032 Marburg, Germany. [22]Université Paris-Saclay, CNRS, AgroParisTech, Ecologie Systématique et Evolution, 91405 Orsay, France. [23]Institute of Forest Ecology, Slovak Academy of Sciences, 96053 Zvolen, Slovakia. [24]Department of Land, Environment, Agriculture and Forestry, University of Padua, Padua, Italy. [25]Office National des Forêts, Département Recherche Développement et Innovation, 77300 Fontainebleau, France. [26]Institute for Atmospheric and Earth System Research/Forest Sciences, Faculty of Agriculture and Forestry, University of Helsinki, 00014 Helsinki, Finland. [27]Swiss Federal Research Institute WSL, Insubric Ecosystems Research Group, 6593 Cadenazzo, Switzerland. [28]Universite de Bordeaux, INRAE, BIOGECO, 33615 Pessac, France. [29]Institute of Botany of the Czech Academy of Sciences, Průhonice, Czech Republic. [30]Department of Botany, Faculty of Science, University of South Bohemia, České Budějovice, Czech Republic. [31]Laboratory for Wood Anatomy and Dendrochronology, Department of Geology, Lund University, Lund, Sweden. [32]Institute of Biology, University of Hohenheim, Stuttgart, Germany. [33]Departamento de Ecología, Centro de Investigaciones sobre Desertificación (CIDE-CSIC), 46113 Moncada Valencia, Spain. [34]National Museum of Natural Sciences, CSIC, 28006 Madrid, Spain. [35]Institute for environmental studies, Faculty of Science, Charles University, Praha, Czech Republic. [36]Department of Wood Science and Technology, Faculty of Forestry and Wood Technology, Mendel University in Brno, 613 00 Brno, Czech Republic. [37]Izmir Katip Çelebi University, Faculty of Forestry, Çigli Izmir, Turkey. [38]Southwest Anatolia Forest Research Institute, Antalya, Turkey. [39]Natural Sciences Unit, German Archaeological Institute, 14195 Berlin, Germany. [40]Brandenburg State Forestry Center of Excellence, Eberswalde, Germany. [41]Chair of Forest Growth and Dendroecology, University of Freiburg, 79085 Freiburg, Germany. [42]Department of Natural Forests, Forest Research Institute, 17-230 Białowieża, Poland. [43]Environmental Measuring Systems Ltd., 621 00 Brno, Czech Republic. [44]Université de Lorraine, AgroParisTech, INRAE, SILVA, F-54000 Nancy, France. [45]Institute for Alpine Environment, Eurac Research, 39100 Bozen/Bolzano, Italy. [46]UCD School of Biology and Environmental Science, University College Dublin, Belfield, Dublin, Ireland. [47]UCD Earth Institute, University College Dublin, Belfield, Dublin, Ireland. [48]Department of Forest Botany, Dendrology and Geobiocoenology, Faculty of Forestry and Wood Technology, Mendel University in Brno, 613 00 Brno, Czech Republic. [49]Department for Geography, University of Bonn, 53115 Bonn, Germany. [50]Institute for Atmospheric and Earth System Research/Physics, Faculty of Science, University of Helsinki, 00014 Helsinki, Finland. [51]Trinity College Dublin, School of Natural Sciences, Botany Department, Dublin, Ireland. [52]National Forest Centre, Forest Research Institute, 96001 Zvolen, Slovakia. [53]Department of Geomatics, Forest Research Institute, 05-090 Raszyn, Poland. [54]Technical University in Zvolen, Faculty of Forestry, 96001 Zvolen, Slovakia. [55]UCD Forestry,

School of Agriculture and Food Science, University College Dublin, Dublin, Ireland. [56]Siberian Federal University, 660041 Krasnoyarsk, Russia. [57]Forestry and Game Management Research Institute, 252 02 Jíloviště, Czech Republic. [58]Southern Swedish Forest Research Centre, Swedish University of Agricultural Sciences (SLU), 230 53 Alnarp, Sweden. [59]Beijing Research & Development Centre for Grass and Environment, Beijing Academy of Agriculture and Forestry Sciences, 100097 Beijing, China. [60]These authors contributed equally: Roberto L. Salomón, Richard L. Peters. ✉email: ute.sassklaassen@wur.nl; kathy.steppe@UGent.be

