## [Peer Review File · Nature Communications]

The 2018 European heatwave led to stem dehydration but not to consistent growth reductions in forestsREVIEWER COMMENTS

Reviewer #1 (Remarks to the Author):

This is a well performed study presenting a unique dataset of dendrometer measurements to assess the impact of the 2018 drought on growth and stem shrinkage across a large number of sites and species in Europe. The main conclusions are that the 2018 heatwave did not reduce growth, but did affect tree water status. Authors further detect a difference between conifers and broadleaf trees in the variation in TWD. The conclusions are well supported by the data and provide a valuable contribution to existing literature that will be of great interest to others in the field.

A major asset of this study is that it is one of the first to apply these dendrometer techniques to a well-replicated spatial network and assess the response to a widespread drought.

I recommend this study for publication, but do have some issues that I feel should be addressed.

1. The article is built on the assumption that stem shrinkage provides a direct measure of stem water status. One can understand that temporary stem shrinkage within a tree can indeed only be due to variation in stem water status, but how should inter-species differences be interpreted? In other words, is there a role for wood structure in causing differences in shrinkage of stems observed between species? Some aspects such as capacitance, elasticity, embolisms resistance are mentioned in the discussion (lines 319-320), but a more elaborate discussion on this issue would be welcome. How does it affect the outcome of this study and could observed differences between conifers and broadleaf trees be explained by their differences in wood structure? This need to be added to the discussion, along with possibly some cautionary notes on the interpretation of these data.

2. If the differences in TWD response between species are indeed purely due to stem water status, can these differences be interpreted in greater depth? The differences between species in TWD are very big (Fig. 2). For most of the species included in this study, information on stomatal control and rooting depth must be available. Could the authors thus try to attribute variation in TWD to stomatal controls (eg. Anisohydric vs isohydric controls), rooting depth, or other traits? Has this been attempted? Why are only 4 species included in the inter-species comparison?

3. I was a bit surprised that the authors did not also analyse TWD anomalies with respect to VPD and REW anomalies. HW2018 did result in greater stem shrinkage but would it not be more powerful to show that magnitude to stem shrinkage is indeed related to the severity of the (soil and atmospheric) drought (relative to sites' background climate)?

4. Finally, there are some methodological issues that require some clarification:

-Lines 583-585 : REW calculations need explaining a bit more. How is the permanent wilting point calculated? Does it vary between sites and soil types? This all needs explaining further. It is not explained why this approach was used instead of using soil moisture water content.

- Do I understand correctly that the second approach regressed the TWD2018:control against the mean VPD and REW within the HW2018 interval (and not the mean VPD and REW for 2016-2017)? This is not made clear in the methods section nor in figure legends and axes (although said at one point in the results). Make this clear throughout the ms, explicitly state this in the methods, and in the figure captions and axes (which should read VPD2018HW , etc., I think).

- Two different approaches were applied for calculating TWD anomalies. In the second approach the 95th percentile of the TWD was taken to calculate the ratio. This is not explained? Please do explain why the approach here differs from approach 1. Also, are the same results obtained when using the mean, or median?

- I may have missed this, but I could not find information on how Figs 2 and 3 were produced. It is a somewhat unusual way of presenting data and does need a bit of explaining. Also, again here, the hydrometeorological space does it represents the background climate or the heatwave REW and VPD?

Minor presentational issues:

-Figure S1: axis of the upper panel is missing. Can this be added?

-Explain in legend what are heatwave extend and heatwave days. Heatwave days are defined as five consecutive days with Tmax daily > 90th percentile. That is clear, but what is heatwave extent? This is not clear. It is said that this when heatwave days overlapped for more than five sites, but that is inconsistent with what is shown in the figure S1.

-Line 208 – 210 : Hydrometeorological space was not different for broadleaf and conifers but the accompanying Figure S3 makes it hard to appreciate this. Can you plot specifically the space for broadleaf next to conifer?

-Line 212-213 : Be more specific. You mean here "... with greater values of TRW2018 corresponding to lower REW and higher VPD during the 2018HW ? Also, do you really mean greater values corresponding to decreasing soil water availability, or simply lower soil water availability (in HW2018)?

-It would be nice if the axes in figs. 3 and 4 could also be added for panels b (at top) and c (left)?

Reviewer #2 (Remarks to the Author):

Salomón et al present an analysis of tree growth and stem water balance response to the extreme drought of 2018 across Europe. Using an innovative approach based on dendrometers and partitioning the dendrometer data between a growth component and a variability component (which is referred to as tree water deficit), they are able to show that temporary shrinkage increased during the drought as a sign of water stress, whereas annual growth seemed rather unaffected. The paper is generally very well written, and the unique and large data sets allows for the generalized conclusions as drawn by the authors, some limitations apply (see below for detailed critique). The concept of the hydrometeorological space is intuitive and helpful, and the associated figures are really great! All findings are interpreted plausibly based on tree ecophysiology. I feel that the statistical analyses are all sound and reproducible. However, I have some doubts concerning the representativity of the network in terms of the regional scope of the HW2018, which might also affect the somewhat controversial finding of no distinct growth response.

Major comments

1) Drought scope and validity of control years

The network of sites has some regional foci in areas that have not been identified as affected by the 2018 drought in other studies. E.g., the sites in central Spain and in Romania would not be classified as being hit by the 2018 drought in e.g. Bastos et al. 2020 and Buras et al. 2020. From figure S1 it also seems that some sites are outside the main area of the 2018 drought impact.

In addition, 2017 was a severe heatwave/drought in Southern Europe, and some of the sites in the network will have experienced more extreme conditions in 2017 in comparison to 2018, see e.g. Rita et al. 2020. This questions the validity of using 2016 and 2017 as control years for all sites. I am wondering if restricting your data set to only those sites where 2018 was the stronger drought than 2017 will yield an even clearer picture of the response - maybe the large spread in TWD (Lines 187ff) could be confined by a more climate-informed selection of the sites?

2) Fixed time frame for drought and control years

If TWD is at least partly controlled by xylem phenology (which I don't know!), then choosing a fixed reference time frame (fixed range of DOYs) could be problematic. Due to different climatic conditions up to this time frame for 2018 (extreme spring warming, only moderate precipitation deficits) in contrast to the control years, trees will have been in a different growth state by the time they reach the reference time frame, which might be associated with a given background level TWD variability. This could easily be ruled out by comparing variability in TWD fluctuations for varying time windows.

3) Immediate growth response in other studies not put in perspective

This study does not find a pronounced impact of the HW2018 on radial growth in the year itself,

which is at odds with reports of immediate growth reduction in association with 2018 (e.g, Rohner et al. 2021 for Switzerland; Larysch et al. 2021 for Germany). Could the absence of a growth response be related to point 1), i.e. site-level effects that cancel each other out?

Minor comment

- L155: HW2018 not defined before here

Reference

Bastos, A., Ciais, P., Friedlingstein, P., Sitch, S., Pongratz, J., Fan, L., Wigneron, J. P., Weber, U., Reichstein, M., Fu, Z., Anthoni, P., Arneth, A., Haverd, V., Jain, A. K., Joetzjer, E., Knauer, J., Lienert, S., Loughran, T., McGuire, P. C., ... Zaehle, S. (2020). Direct and seasonal legacy effects of the 2018 heat wave and drought on European ecosystem productivity. *Science Advances*, 6(24), eaba2724. <https://doi.org/10.1126/sciadv.aba2724>

Buras, A., Rammig, A., & Zang, C. S. (2020). Quantifying impacts of the 2018 drought on European ecosystems in comparison to 2003. *Biogeosciences*, 17, 1655–1672.

Larysch, E., Stangler, D. F., Nazari, M., Seifert, T., & Kahle, H.-P. (2021). Xylem Phenology and Growth Response of European Beech, Silver Fir and Scots Pine along an Elevational Gradient during the Extreme Drought Year 2018. *Forests*, 12(1), 75. <https://doi.org/10.3390/f12010075>

Rita, A., Camarero, J. J., Nolè, A., Borghetti, M., Brunetti, M., Pergola, N., Serio, C., Vicente-Serrano, S. M., Tramutoli, V., & Ripullone, F. (2020). The impact of drought spells on forests depends on site conditions: The case of 2017 summer heat wave in southern Europe. *Global Change Biology*, 26(2), 851–863. <https://doi.org/10.1111/gcb.14825>

Rohner, B., Kumar, S., Liechti, K., Gessler, A., & Ferretti, M. (2021). Tree vitality indicators revealed a rapid response of beech forests to the 2018 drought. *Ecological Indicators*, 120, 106903. <https://doi.org/10.1016/j.ecolind.2020.106903>

Response to reviewers letter | Manuscript "The 2018 European heatwave led to stem dehydration but not to consistent growth reductions" (NCOMMS-21-21559)

Reviewer #1

This is a well performed study presenting a unique dataset of dendrometer measurements to assess the impact of the 2018 drought on growth and stem shrinkage across a large number of sites and species in Europe. The main conclusions are that the 2018 heatwave did not reduce growth, but did affect tree water status. Authors further detect a difference between conifers and broadleaf trees in the variation in TWD. The conclusions are well supported by the data and provide a valuable contribution to existing literature that will be of great interest to others in the field. A major asset of this study is that it is one of the first to apply these dendrometer techniques to a well-replicated spatial network and assess the response to a widespread drought.

I recommend this study for publication, but do have some issues that I feel should be addressed.

Response: We thank the reviewer for the positive assessment of our manuscript. We carefully went through the raised issues, performed additional analyses, and adjusted the text accordingly. With these considerations, we hope we have addressed the reviewers' concerns.

1. The article is built on the assumption that stem shrinkage provides a direct measure of stem water status. One can understand that temporary stem shrinkage within a tree can indeed only be due to variation in stem water status, but how should inter-species differences be interpreted? In other words, is there a role for wood structure in causing differences in shrinkage of stems observed between species? Some aspects such as capacitance, elasticity, embolisms resistance are mentioned in the discussion (lines 319-320), but a more elaborate discussion on this issue would be welcome. How does it affect the outcome of this study and could observed differences between conifers and broadleaf trees be explained by their differences in wood structure? This needs to be added to the discussion, along with possibly some cautionary notes on the interpretation of these data.

Response: Differences in wood structure, xylem resistance to embolism, hydraulic capacitance, and elasticity indeed affect the capacity of stem water depletion and refilling of a tree species, which overall determines water-related stem diameter fluctuations. Therefore, we used tree-specific response ratios for the analyses and comparisons to account for variability related to individual stem size, bark thickness, and abovementioned wood traits. For this, we estimated tree-specific ratios of GRO and TWD in 2018 divided by control years (2016 and 2017). These tree-specific ratios allow for a more straightforward comparison of stem shrinkage during HW2018 in relation to the control period among species. We acknowledge that this key point was not clear enough and it has now been emphasized earlier in the revised text (at the end of the introduction; L194-200): "Tree-specific normalization of 2018 data relative to control years (see Methods) accounted for variability related to individual stem size, bark thickness, local environmental conditions, and wood traits that might affect depletion and refilling rates of stem water reserves, such as xylem hydraulic conductivity, hydraulic capacitance, wood elasticity, and xylem resistance to embolism formation. Tree-specific response ratios therefore allow for comparison of TWD and GRO across species and sites, highlighting differential temporal dynamics due to HW2018".

As noted by the reviewer, absolute values of TWD and GRO were indeed largely affected by wood traits, as denoted by the large variability among species (Fig. S3). Particularly the hydraulic resistances the wood exerts on axial and radial water flow impact the water potential gradients generated within tree stems and therefore the rates of depletion and refilling of stem water reservoirs. For instance, highly conductive wood structures, like ring-porous species (i.e., *Quercus robur*¹), commonly show a lower axial hydraulic resistance and thus would have greater ease at refilling the storage tissues when water is available and atmospheric demand decreases. Coniferous species commonly show a higher hydraulic resistance² and would thus be more strained at rehydrating their storage tissues when

atmospheric water demand decreases. Moreover, although not assessed with dendrometers, the wood anatomical structure, tissue elasticity, and the presence and abundance of radial pathways for water flow also affect the hydraulic capacitance of sapwood and bark tissues. Species with larger hydraulic capacitance can buffer sudden increases in xylem tension providing xylem hydraulic safety^{3,4} and would be expected to experience greater shrinkage (and hence TWD). These relevant points should encourage parallel research, as noted and extended along the discussion in the revised manuscript (L338-344). However, in this study we focus on performing an observational study on the impact of the 2018 heatwave, where explanatory mechanisms are beyond the scope of this study, as no information is available on the wood traits of the specific monitored tree stems. Additionally, destructive sampling is not advised when continuously monitoring a tree over many years.

2. If the differences in TWD response between species are indeed purely due to stem water status, can these differences be interpreted in greater depth? The differences between species in TWD are very big (Fig. 2). For most of the species included in this study, information on stomatal control and rooting depth must be available. Could the authors thus try to attribute variation in TWD to stomatal controls (eg. Anisohydric vs isohydric controls), rooting depth, or other traits? Has this been attempted? Why are only 4 species included in the inter-species comparison?

Response: With respect to differences in TWD responses, we should keep in mind that response ratios to HW2018 were compared among species and not absolute values. Finding physiological links between the absolute stem water status and other hydraulic traits and mechanisms is indeed of great interest. We are aware of multiple trait databases which could be used for such analyses, including root traits⁵, general plant hydraulic traits⁶⁻⁸ and leaf specific traits^{9,10}. Moreover, inter-specific comparisons considering sap flow data and traits have certainly already been performed¹¹. The limitation on performing such analyses in this work is rooted in the fact that we are constrained by the available dendrometer data, collected from 2016 till 2018, which narrows the number of species and thus precludes finding sufficient species-specific and spatial overlap between globally collected traits and our compiled data for sound analyses.

We have attempted to link the TWD response ratio to species-specific root traits (see Fig. R1 in this response letter), as the often unknown root architecture plays a critical role in facilitating water access. Although rooting databases are available⁵, linking this information to our tree-centered dataset was unfeasible as the available data is quite sparsely distributed and the analysis outcome would be highly uncertain. We do, however, agree that this would be an important aspect to explore in the future and have thus added this consideration to the discussion (L338-344): "Other hydraulic traits are likely involved in the complex species-specific response of stem water reserves to heatwaves. For instance, we could expect that tree species with higher stem hydraulic capacitance and less dense wood, and hence a lower Young's modulus of elasticity, would experience larger stem shrinkage and swelling for a given change in xylem tension. Linking our dendrometer findings to leaf, root and plant specific hydraulic traits would be a critical next step, yet requires a more intense data collection in species richness and spatial detail".

Fig. R1. Example of meta-analysis on root traits of common European tree species. Data originates from the GRoot database⁵. Target species are highlighted in the legend, indicated with consistent colors. *Quercus spp.* includes *Q. robur* and *Q. petraea*. (a) Sampling locations for root traits in Europe. (b) Rooting depth recorded for the target species. (c) Root production rate for the target species. For (b) and (c) both the maximum range of data is highlighted (transparent polygon) and 5th and 95th percentile (solid polygon).

The inter-specific comparison was performed on the four most important forest species of Europe, both in terms of spatial distribution and economic importance¹² (see Methods and results, L241-243, 704-707). This analysis was uniquely performed on these four species because the number of sites was quite low for the other species within the available dataset (Fig. R2) and therefore did not cover broad gradients of climatic conditions (in terms of VPD and REW). As the most dominant species within the database are *Fagus sylvatica*, *Picea abies*, *Quercus robur/petraea* and *Pinus sylvestris* (Fig. R2) we focused our mixed-effect modelling efforts on these species. This consideration has been added to the method section to further clarify this point (L708-710): “Moreover, these species were selected due to their appropriate spatial coverage within the available dataset, both in terms of number of sites and registered gradients of climatic conditions.”

Fig. R2. Number of sites per species available within the database.

3. I was a bit surprised that the authors did not also analyze TWD anomalies with respect to VPD and REW anomalies. HW2018 did result in greater stem shrinkage but would it not be more powerful to show that magnitude to stem shrinkage is indeed related to the severity of the (soil and atmospheric) drought (relative to sites' background climate)?

Response: We thank the reviewer for making this relevant statement. When performing the analyses we already discussed this crucial point with the co-authors and decided for the analyses shown in Figs. 3 and 4 to compare TWD response ratios across comparable climatic conditions in 2018 – in absolute values of vapor pressure deficit (VPD) and relative extractable water (REW) – as described in the methods (L690-694). As such, we accounted for the absolute variability in heatwave climatic conditions at the different sites. Moreover, as we solely have 2016 and 2017 as control years for the TWD data, we were not able to establish the long-term mean shrinkage patterns observed for the species, which would be unbalanced in comparison to the available data regarding long-term background climatic conditions. Therefore, we decided to maintain the analyses as it is within the manuscript. However, we do agree that it might be relevant to test whether stem shrinkage patterns are similar when accounting for the background site climate in terms of VPD and REW. We performed similar analyses as presented in Fig. 3 (absolute VPD and REW as explanatory variables) with relative VPD and REW as explanatory variables. For this, the 2018 REW and VPD conditions during the heatwave period (day of year 208 till 264) were divided by the mean REW and VPD conditions during the heatwave period from 2000 till 2017 (considered as the site's background climate; Fig. R3). Values <1 for the ratio of REW ($REW_{2018:control}$) indicate drier soil conditions compared to the background conditions, while values >1 for the ratio of VPD ($VPD_{2018:control}$) indicate periods with higher atmospheric demand compared to the background conditions. We have incorporated this Figure within Supplementary Information (as Fig. S6) and refer to it along the results (L233-240). These additional analyses confirm the general response of greater stem shrinkage with increasing relative VPD and decreasing REW. Moreover, it verifies that the main difference in shrinkage patterns is found for the daily minimum TWD, where conifers show higher $TWD_{2018:control}$ ratios over a larger (relative and absolute) hydrometeorological space area. However, due to the lower overlap in the common hydrometeorological space area, the proportion of daily min. $TWD > 1$ for broadleaved species increased.

Fig. R3. Response of tree water deficit (TWD) to the relative change in vapor pressure deficit ($VPD_{2018:control}$) and relative extractable water ($REW_{2018:control}$) of broadleaf (a, c) and conifer (b, d) species. The hydrometeorological space was determined by dividing the daily 2018 VPD and REW values during the heatwave period (day of year 208 till 264) by the mean VPD and REW conditions during the same heatwave period from 2000 till 2017. a-d, Linear-mixed effect model output of the ratio of the daily minimum (a, b) and maximum (c, d) TWD in the 2018 heatwave compared to the 95th percentile of the control period ($TWD_{2018:control}$). $TWD_{2018:control}$ above 1 indicates a larger shrinkage was registered during the 2018 heatwave compared to the control period. From the hydrometeorological space range of the entire database (indicated in grey), models have been adjusted for the common climatic range of broadleaf and conifer species only (indicated with dotted lines). e, Bars indicate the percentage of the hydrometeorological space covering different ranges of $TWD_{2018:control}$ values (see legend).

4. Finally, there are some methodological issues that require some clarification:

-Lines 583-585: REW calculations need explaining a bit more. How is the permanent wilting point calculated? Does it vary between sites and soil types? This all needs explaining further. It is not explained why this approach was used instead of using soil moisture water content.

Response: We determined field capacity and permanent wilting point to standardize the data to relative extractable water (REW; see methods L639-640). Determining REW is often done in meta-analyses when information on the soil texture class and bulk density is lacking, which calls for the use of soil water potential¹³, likely a better indicator of plant drought stress, as this is independent from soil properties¹⁴. Yet, within our study neither soil water potential data nor detailed soil descriptions at the site level were available. Moreover, relating local soil texture class measurements to the used larger scaled-products could introduce uncertainty due to local soil heterogeneity. As such, we could solely work with the raw soil moisture (SM) values, which can lead to equivocal conclusions when comparing multiple sites with different soil texture classes as the points of near water saturation (field capacity; blue line in Fig. R4) and highest soil drainage (orange line in Fig. R4) can substantially vary (Fig. R5).

Fig. R4. Example of soil moisture content time-series for a site in the Czech Republic (left panel) and Austria (right panel; see Table S1). Relative extractable water (REW) is determined by using the 95th percentile and the site-specific minimum.

We thus calculated the commonly applied REW, a normalized measure of soil water moisture^{11,15}. The standard normalization was performed according to:

$$REW_{j,i} = \frac{SM_{j,i} - SM_{min,j}}{SM_{95th,j} - SM_{min,j}}$$

where $REW_{j,i}$ and $SM_{j,i}$ are site (j) and daily (i) values, and $SM_{95th,j}$ and SM_{min} are the points of approaching field capacity and minimum SM values, respectively. The 95th percentile, and not the maximum value, was used to avoid including rain events which cause full soil water saturation (i.e., Fig. R4). We realized, thanks to the reviewer’s comment, that the soils likely do not reach permanent wilting point during the survey period here. We incorrectly named this minimum value as such, and should refer to it as minimum SM recorded in the series. Although the minimum SM is arguable a less standardized metric, we think it is the optimal practice when having limited information on site-specific soil conditions. Following these considerations, we have rephrased this point in the text (L639-643): “Relative extractable water (REW) was calculated to account for inter-site differences in soil properties and absolute soil moisture levels, a commonly used normalized metric of soil water availability to facilitate inter-site comparisons⁵⁹. REW was calculated by scaling the site-specific daily soil moisture simulations to field capacity (95th upper percentile = 1, hereby excluding rain events that may result in soil water saturation) and the site lowest soil moisture value (minimum value = 0)”.

The referee correctly notified that there is variability in both field capacity values and sites’ absolute minimum (Fig. S5). Unfortunately, we do not have site-specific soil information which could explain these differences and further refine proxies of available soil water.

Fig. R5. Variability in field capacity (95th percentile) and the site-specific minimum soil moisture.

- Do I understand correctly that the second approach regressed the TWD2018:control against the mean VPD and REW within the HW2018 interval (and not the mean VPD and REW for 2016-2017)? This is not made clear in the methods section nor in figure legends and axes (although said at one point in the results). Make this clear throughout the ms, explicitly state this in the methods, and in the figure captions and axes (which should read VPD2018HW, etc., I think).

Response: This is a correct point raised by the referee and connects to point#3. We indeed selected climatic conditions experienced during HW2018 (absolute VPD and REW) for analyses, while standardizing TWD for HW2018 to the control period (2016-2017) as explained above (point#1). This approach was used to ensure that similar climatic conditions in 2018 were compared for the tree-specific daily TWD response ratios from the different sites. We adjusted the methods and the text throughout the manuscript to clearly mention this point (L194-200, 215-218, 690-694). Moreover, axes in Figs. 3 and 4 have been adjusted accordingly to further clarify that we refer to the hydrometeorological space in absolute terms during HW2018.

- Two different approaches were applied for calculating TWD anomalies. In the second approach the 95th percentile of the TWD was taken to calculate the ratio. This is not explained? Please do explain why the approach here differs from approach 1. Also, are the same results obtained when using the mean, or median?

Response: We thank the referee for pointing us to this unclear description of the methods. We state for the second approach that “the ratio between the 2018 daily TWD and the 95th percentile of the control years (2016-2017) within the heatwave timeframe was calculated” (L694-695). To understand this difference, we need to consider the time scale of the two approaches. The first approach works with annual values, while the second approach uses daily values. To avoid confusion, we adjusted the text (L690-691): “In contrast to the annual values used in the first approach, the second approach assessed the daily response of minimum and maximum TWD to HW2018 (Figs. 3-4, Tables S3-S4).” For the daily analyses (second approach) we decided to use the 95th percentile of TWD within the control period (2016-2017) instead of the mean as this metric informs about the maximum shrinkage (avoiding outliers) experienced by the tree before the 2018 heatwave. The reasoning behind this is that one can compare the daily shrinkage during HW2018 to the maximum shrinkage experienced by the tree during the control period focusing on the daily intensity of stem shrinkage. Note that for the first approach (annual time scale – Fig. 2), comparisons were made using mean values to better compare TWD in terms of both intensity and duration.

Indeed, we also considered using the mean or the median value for the second approach. An example of using median and 95th percentile values for two dendrometer series is shown in Fig. R6. The issue with using the median is apparent from this figure, where the proportion of days with fully refilled (hydrated) stems (i.e., TWD = 0) during the HW2018 time frame result in low median (or mean) values

(upper panels) and subsequently generates disproportionately high TWD daily ratios for the daily values (lower panels) as seen in the FRFon-site. Due to the heterogeneity in number of days with full refilling across sites, this makes obtaining appropriate model fits for the hydrometeorological space of 2018 implausible, which is solved when using the 95th percentile values. As such, we decided to go for the 95th percentile and not present results using median (or mean) values. This point has been clarified in methods (L695-699).

Fig. R6. Tree water deficit values and their subsequent calculated ratios for a tree growing in the Czech Republic (left hand panels) and France (right hand panels). Within the upper panels both the 95th percentile (blue line) and the median (cyan line) values were calculated through days covering the heatwave period (orange dots) during the control period (2016-2017). Within the two lower panels the subsequent ratios to the control period values are plotted. Mind that some tree water deficit peaks in winter are caused by well-known winter shrinkage patterns.

- I may have missed this, but I could not find information on how Figs 2 and 3 were produced. It is a somewhat unusual way of presenting data and does need a bit of explaining. Also, again here, the hydrometeorological space does it represent the background climate or the heatwave REW and VPD?

Response: We believe the reviewer refers to Figs. 3 and 4. These figures show the resulting modelling fits of the climatic response analysis, where linear mixed effect models were fitted to predict min. and max. daily $TWD_{2018:control}$ as a function of daily VPD (using a 2nd order polynomial) and REW (using a 3rd order polynomial) of 2018, with tree nested within site, and species as crossed random (intercept) effects. The models were used to generate a 3-dimensional figure presenting the TWD ratio of 2018 to the control period ($TWD_{2018:control}$) for each daily VPD and REW condition during HW2018. We agree that presenting model outputs in a 3-dimensional space is not common (but see reference¹⁴). We thus clarified the interpretation of this figure in the text (L218-221): “This procedure was used to ensure the comparison of daily minimum and maximum $TWD_{2018:control}$ under comparable climatic conditions, and is presented in a 3-dimensional space using a linear-mixed effect model with a polynomial structure (see Methods).”

Moreover, the hydrometeorological space represent the absolute REW and VPD during 2018, which has been clarified in the figure caption and axis titles.

Minor presentational issues:

-Figure S1: axis of the upper panel is missing. Can this be added?

Response: We thank the referee for this detailed consideration. We included the axis in the upper panel as requested (Extended Material Fig. S1).

Fig. S1 | Heatwave extent in 2018 for all sites included within the network. a-b, Frequency distribution and site-specific distribution of the heatwave extent and days. A heatwave is defined as five consecutive days with a maximum daily temperature higher than the 90th percentile of the control period (1951-2000 E-OBS data). The selected heatwave extent for analyses is indicated with dotted lines (from DOY 208 until 264). This extent shows the day of year (DOY) when more than five sites overlap in heatwave days to define the start and the end of the heatwave period.

-Explain in legend what are heatwave extend and heatwave days. Heatwave days are defined as five consecutive days with Tmax daily > 90th percentile. That is clear, but what is heatwave extent? This is not clear. It is said that this is when heatwave days overlapped for more than five sites, but that is inconsistent with what is shown in the figure S1.

Response: We adjusted the figure caption accordingly (see point above). We clarified the selection of the dotted lines (heatwave period) as the day of year (DOY) when more than five sites show an overlap in heatwave days to define the start and the end of the heatwave period. Within the figure this is given by the DOY when more than five grey rectangles are overlapping. In panel (a), this is clear at the point where the polygon goes above five.

-Line 208 – 210 : Hydrometeorological space was not different for broadleaf and conifers but the accompanying Figure S3 makes it hard to appreciate this. Can you plot specifically the space for broadleaf next to conifer?

Response: We agree that for the analyses presented in Fig. 3 one needs to get an idea on the actual climate space covered within the analyses. As such we incorporated Fig. R7 within the supporting information (as Fig. S5) and the text has been adapted accordingly (L232-235).

Fig. R7| Hydrometeorological space for conifer and broadleaf species. Site-specific daily mean vapor pressure deficit (VPD) and relative extractable water (REW) per taxonomic clade during the 2018 European heatwave (from DOY 208 until 264). Darker point clouds indicate a higher occurrence of days with similar VPD and REW conditions.

-Line 212-213 : Be more specific. You mean here "... with greater values of TRW2018 corresponding to lower REW and higher VPD during the 2018HW? Also, do you really mean greater values corresponding to decreasing soil water availability, or simply lower soil water availability (in HW2018)?

Response: We thank the reviewer for pointing us to this unclear description of the results. We adjusted the text accordingly, which now reads (L221-224): "Both the daily minimum and maximum $TWD_{2018:control}$ showed significant responses to decreasing REW and increasing VPD for broadleaf and conifer species ($P < 0.05$; Table S3), with greater $TWD_{2018:control}$ ratios during periods of lower soil water availability and higher atmospheric evaporative demand (Fig. 3a-d)."

-It would be nice if the axes in figs. 3 and 4 could also be added for panels b (at top) and c (left)?

Response: We added the axes as suggested by the reviewer (see new Figs. 3 and 4).

Fig. 3 | Response of tree water deficit (TWD) to vapor pressure deficit (VPD, kPa) and relative extractable water (REW, unitless [-]) of broadleaf (a, c) and conifer (b, d) species in the VPD-REW climate space.

Fig. 4 | Species-specific response of tree water deficit (TWD) to vapor pressure deficit (VPD) and relative extractable water (REW) in the VPD-REW climate space.

References in response letter (Reviewer #1)

1. Steppe, K. & Lemeur, R. Effects of ring-porous and diffuse-porous stem wood anatomy on the hydraulic parameters used in a water flow and storage model. *Tree Physiol.* **27**, 43–52 (2007).
2. Tyree, M. & Ewers, F. W. The hydraulic architecture of trees and other woody plants. *New Phytol.* **119**, 345–360 (1991).
3. Meinzer, F. C., Johnson, D. M., Lachenbruch, B., McCulloh, K. A. & Woodruff, D. R. Xylem hydraulic safety margins in woody plants: coordination of stomatal control of xylem tension with hydraulic capacitance. *Funct. Ecol.* **23**, 922–930 (2009).
4. McCulloh, K. A., Johnson, D. M., Meinzer, F. C. & Woodruff, D. R. The dynamic pipeline: hydraulic capacitance and xylem hydraulic safety in four tall conifer species. *Plant. Cell Environ.* **37**, 1171–1183 (2014).
5. Guerrero-Ramírez, N. R. *et al.* Global root traits (GRoot) database. *Glob. Ecol. Biogeogr.* **30**, 25–37 (2021).
6. Mencuccini, M. *et al.* Leaf economics and plant hydraulics drive leaf : wood area ratios. *New Phytol.* **224**, 1544–1556 (2019).
7. Sanchez-Martinez, P., Martínez-Vilalta, J., Dexter, K. G., Segovia, R. A. & Mencuccini, M. Adaptation and coordinated evolution of plant hydraulic traits. *Ecol. Lett.* **23**, 1599–1610 (2020).
8. Kattge, J. *et al.* TRY plant trait database – enhanced coverage and open access. *Glob. Chang. Biol.* **26**, 119–188 (2020).
9. Wright, I. J. *et al.* The worldwide leaf economics spectrum. *Nature* **428**, 821–827 (2004).
10. Wright, I. J. *et al.* Global climatic drivers of leaf size. *Science (80-.)*. **357**, 917–921 (2017).
11. Flo, V. *et al.* Climate and functional traits jointly mediate tree water-use strategies. *New Phytol.* (2021). doi:10.1111/nph.17404
12. Hanewinkel, M., Cullmann, D. A., Schelhaas, M.-J., Nabuurs, G.-J. & Zimmermann, N. E. Climate change may cause severe loss in the economic value of European forest land. *Nat. Clim. Chang.* **3**, 203–207 (2013).
13. Teepe, R., Dilling, H. & Beese, F. Estimating water retention curves of forest soils from soil texture and bulk density. *J. Plant Nutr. Soil Sci.* **166**, 111–119 (2003).
14. Zweifel, R. *et al.* Why trees grow at night. *New Phytol.* nph.17552 (2021). doi:10.1111/nph.17552
15. Granier, A. *et al.* Evidence for soil water control on carbon and water dynamics in European forests during the extremely dry year: 2003. *Agric. For. Meteorol.* **143**, 123–145 (2007).

Reviewer #2

Salomón et al present an analysis of tree growth and stem water balance response to the extreme drought of 2018 across Europe. Using an innovative approach based on dendrometers and partitioning the dendrometer data between a growth component and a variability component (which is referred to as tree water deficit), they are able to show that temporary shrinkage increased during the drought as a sign of water stress, whereas annual growth seemed rather unaffected. The paper is generally very well written, and the unique and large data sets allows for the generalized conclusions as drawn by the authors, some limitations apply (see below for detailed critique). The concept of the hydrometeorological space is intuitive and helpful, and the associated figures are really great! All findings are interpreted plausibly based on tree ecophysiology. I feel that the statistical analyses are all sound and reproducible. However, I have some doubts concerning the representativity of the network in terms of the regional scope of the HW2018, which might also affect the somewhat controversial finding of no distinct growth response.

Response: We thank the reviewer for the very positive assessment of our manuscript, particularly in relation to the performed analyses and data representation. The raised concerns are relevant and aided us in further improving the work. We performed additional analyses to tackle these issues and welcomed the suggested literature for consideration. We hope to have addressed the limitations postulated by the reviewer.

Major comments

1) Drought scope and validity of control years

The network of sites has some regional foci in areas that have not been identified as affected by the 2018 drought in other studies. E.g., the sites in central Spain and in Romania would not be classified as being hit by the 2018 drought in e.g. Bastos et al. 2020 and Buras et al. 2020. From Fig. S1 it also seems that some sites are outside the main area of the 2018 drought impact.

In addition, 2017 was a severe heatwave/drought in Southern Europe, and some of the sites in the network will have experienced more extreme conditions in 2017 in comparison to 2018, see e.g. Rita et al. 2020. This questions the validity of using 2016 and 2017 as control years for all sites. I am wondering if restricting your data set to only those sites where 2018 was the stronger drought than 2017 will yield an even clearer picture of the response - maybe the large spread in TWD (Lines 187ff) could be confined by a more climate-informed selection of the sites?

Response: We thank the referee for this relevant comment. Indeed, the 2018 European heatwave showed substantial temporal and spatial variability. Although we were not able to incorporate sites from central Spain in the dataset, we do have sites from Romania and Southern France (Fig. 1), for which environmental stress might have not particularly threatening during summer 2018. When comparing both the average VPD and REW during the HW2018 period with the corresponding period in 2017, it is indeed clear that the Romanian sites in particular did not experience higher VPD and lower REW during 2018 (Fig. R8).

Fig. R8 | Climate-informed selection of sites. Site-specific values on the difference between 2018 and 2017 vapor pressure deficit (VPD) and relative extractable water (REW). For both years the mean values during the heatwave period (day of year 208 till 265) were determined, after which the difference (Δ) was calculated. Sites with lower VPD or higher REW in 2018 compared to 2017 are highlighted in orange.

We agree with the suggestion of the reviewer to test restricting the data to those sites significantly hit by the HW2018 and assess whether results, especially the lack of a growth response, remain the same. It should be noted that it is unclear how the reviewer defines a stronger drought, as this can be atmospheric (VPD) or soil (REW) driven. This difficulty becomes apparent in Fig. R8, where several sites show higher VPD during 2018 while not showing lower REW or vice versa. We re-run the analysis presented in Fig. 2 (considering 175 trees from 37 sites) by selecting sites with higher VPD and lower REW during the 2018 heatwave period compared to 2017 (black dots in Fig. R8). According to this criterion, excluded sites were mainly distributed across Eastern Europe (i.e., Czech Republic, Romania) and the dataset was reduced to 150 trees from 30 sites. When re-running the analyses (Fig. R9), minimum and maximum $TWD_{2018:control}$ results did not significantly change in relation to those reported in the main text (TWD in 2018 was greater than during the control years, and similar for both taxonomic clades). Results of the $GRO_{2018:control}$ ratio did slightly change in relation to the previous analyses (with less strict site-selection criteria): a non-significant trend ($P = 0.06$) of lower growth during 2018 compared to control years was noticed, with no significant differences between conifers and broadleaves. This new observation has been included and discussed along the text (L178-181, L209-212, 257-262), and Fig. R8 above has been also included in the Supplementary Information (as Fig. S2).

Fig. R9 | Species-specific ratios of tree water deficit (TWD) during the 2018 heatwave and 2018 annual growth (GRO) relative to control years (2016-2017). Solely sites were included within these analyses where vapor pressure deficit and relative extractable water were higher and lower in the heatwave periods of 2018 compared to 2017, respectively. Graphical presentation is similar to that of Fig. 2.

Thanks to the suggestion of the referee, while performing this new analysis, we found a unit issue which affected absolute values of TWD and GRO; for a few sites, data was not converted into μm in the radial direction (as for most of the sites). This issue has now been solved. Absolute values of GRO and TWD are obviously different, and Fig. S3 has been corrected accordingly. The issue does not affect log transformed $\text{TWD}_{2018:\text{control}}$ ratios (nor statistical analyses and main conclusion) as unit differences equally affected the numerator and denominator of the response ratio. However, the unit addition ($1 \mu\text{m}$) to GRO before log transformation (to maintain trees with null growth within analyses) did slightly affected $\text{GRO}_{2018:\text{control}}$ ratios, but not the main conclusions derived from statistical analyses. Fig. 2b, Table S2 and results (L207-209) have been adapted accordingly.

2) Fixed time frame for drought and control years

If TWD is at least partly controlled by xylem phenology (which I don't know!), then choosing a fixed reference time frame (fixed range of DOYs) could be problematic. Due to different climatic conditions up to this time frame for 2018 (extreme spring warming, only moderate precipitation deficits) in contrast to the control years, trees will have been in a different growth state by the time they reach the reference time frame, which might be associated with a given background level TWD variability. This could easily be ruled out by comparing variability in TWD fluctuations for varying time windows.

Response: Xylem phenology exerts a limited control on TWD since the wood is generally more rigid than the bark. TWD dynamics mainly reflect bark tissue shrinkage and swelling (L147-151) due to use and replenishment of internally stored water in elastic bark tissues. While critical questions remain on how phloem physiology (as part of the bark) changes seasonally¹, early studies report that phloem cells are produced before new xylem differentiation begins, and that phloem differentiation precedes xylem differentiation by approximately $1 \frac{1}{2}$ months². This suggests that bark internal water reservoirs are mainly present and functional from early season onwards. Moreover, changes in bark water content due to the flow of water from bark to xylem, resulting in bark tissue shrinkage measured with dendrometers, were demonstrated with magnetic resonance imaging (MRI)³. As such, TWD is a useful metric to quantify the relative importance of internally stored water during drought.

As detailed in point #1 (reviewer #1), standardized tree-specific ratios were used in our study to allow for a straightforward comparison of stem shrinkage during HW2018 in relation to the control period among species. Given that TWD reflects the instant use of internally stored water in the present bark, the chosen fixed reference time frame (from day of year 208 until 264) can be used without problem. This instant use of water and the ability to measure it, point to the early-warning applications that resides in dendrometer measurements.

If trees were in a different growth stage due to different climatic conditions up to the HW2018 time frame, the existing internal water reservoirs are expected to be relatively similar, further indicating that standardized tree-specific ratios with fixed time frames can be used. Moreover, TWD is estimated as stem shrinkage relative to the preceding maximum (Fig. 1c), so TWD during periods of limited drought stress (before HW2018 time frame) would tend to be zero and seasonal TWD averages during 2018 would therefore decrease, potentially leading to $TWD_{2018:control}$ ratios approaching one. Enlarging the time window (or using different time windows during periods of limited drought stress) might therefore hinder our ability to detect differences in TWD patterns during HW2018 in relation to the control period (Figure 2), i.e., as the differences will be smaller. For daily analyses (Figures 3 and 4), including days with limited drought stress might lead to a larger number of points with relatively low TWD ratios, but this would not change the output of polynomial models and the main results, as we used for daily analyses the 95th percentile of TWD during the control period.

3) Immediate growth response in other studies not put in perspective

This study does not find a pronounced impact of the HW2018 on radial growth in the year itself, which is at odds with reports of immediate growth reduction in association with 2018 (e.g, Rohner et al. 2021 for Switzerland; Larysch et al. 2021 for Germany). Could the absence of a growth response be related to point 1), i.e. site-level effects that cancel each other out?

Response: We appreciate this comment and realized that the overall neutral response of stem growth to HW2018 was weakly discussed. Stem growth reductions in 2018 have been indeed observed in some regional studies (e.g., Rohner et al. 2021; Larysch et al. 2021), and also here at the species level (see species-specific boxes in Fig. 2b below the zero dashed line). In contrast, other regional studies have reported a neutral response, and as noted along point#1, a restrictive criteria of site selection considering uniquely those sites where HW2018 effectively led to greater soil and atmospheric drought stress resulted in slightly different results (a non-significant trend of growth reduction; $P = 0.06$). We therefore agree with the referee that a variable $GRO_{2018:control}$ response among sites and species are able to cancel out each other.

This important observation has been included in detail to open the discussion (L254-262): “Our results partially supported hypothesis 1, as stems experienced larger shrinkage during HW2018 relative to control years. However, no consistent reductions in annual stem growth were found (Fig. 2), as a variable response among sites and species cancelled each other out. Illustratively, regional studies have reported negative^{28,29}, neutral^{30,31}, and positive³² responses of annual stem growth to HW2018. Similarly, our results were dependent on the site selection for analysis, as more restrictive criteria according to HW2018 intensity tended to increase the plausibility of growth reductions. Limitations to stem-girth increment are therefore highly dependent on site-specific conditions and, importantly, on the timing of the extreme climatic events. [...]”

Minor comment

- L155: HW2018 not defined before here

Response: The 2018 heatwave (HW2018) has been defined now on its first mention (L157-158).

Reference given by reviewer #2:

Bastos, A., Ciais, P., Friedlingstein, P., Sitch, S., Pongratz, J., Fan, L., Wigneron, J. P., Weber, U., Reichstein, M., Fu, Z., Anthoni, P., Arneth, A., Haverd, V., Jain, A. K., Joetzjer, E., Knauer, J., Lienert, S., Loughran, T., McGuire, P. C., ... Zaehle, S. (2020). Direct and seasonal legacy effects of the 2018 heat wave and drought on European ecosystem productivity. *Science Advances*, 6(24), eaba2724.
<https://doi.org/10.1126/sciadv.aba2724>

Buras, A., Rammig, A., & Zang, C. S. (2020). Quantifying impacts of the 2018 drought on European ecosystems in comparison to 2003. *Biogeosciences*, 17, 1655–1672.

Larysch, E., Stangler, D. F., Nazari, M., Seifert, T., & Kahle, H.-P. (2021). Xylem Phenology and Growth Response of European Beech, Silver Fir and Scots Pine along an Elevational Gradient during the Extreme Drought Year 2018. *Forests*, 12(1), 75.

Rita, A., Camarero, J. J., Nolè, A., Borghetti, M., Brunetti, M., Pergola, N., Serio, C., Vicente-Serrano, S. M., Tramutoli, V., & Ripullone, F. (2020). The impact of drought spells on forests depends on site conditions: The case of 2017 summer heat wave in southern Europe. *Global Change Biology*, 26(2), 851–863.
<https://doi.org/10.1111/gcb.14825>

Rohner, B., Kumar, S., Liechti, K., Gessler, A., & Ferretti, M. (2021). Tree vitality indicators revealed a rapid response of beech forests to the 2018 drought. *Ecological Indicators*, 120, 106903. <https://doi.org/10.1016/j.ecolind.2020.106903>

References in response letter (Reviewer #2)

1. Savage, J.A. It's all about timing—or is it? Exploring the potential connection between phloem physiology and whole plant phenology. *American Journal of Botany* 107(6): 848–851 (2020).
2. Alfieri, F.J. & Evert, R.F. Structure and seasonal development of secondary phloem in Pinaceae. *Botanical Gazette* 134: 17-25 (1973).
3. De Schepper, V., van Dusschoten, D., Copini, P., Jahnke, S., Steppe, K. MRI links stem water content to stem diameter variations in transpiring trees. *Journal of Experimental Botany* 63, 2645-2653 (2012).

REVIEWERS' COMMENTS

Reviewer #1 (Remarks to the Author):

The authors responded exceptionally well to my comments and I am happy to recommend publication of this manuscript.

Perhaps there are three small points that could still be considered.

Firstly, I find the fig. R4 and the formula for REW highly illustrative and wonder if this could be included in the ms methods (both the equation and explanation of the symbols, and the figure). Of course the reader can also look up the cited references but it would make it easier for the reader if it is included in the text.

Secondly, should the added text in line 343/344 not read " .. yet requires data for more species with greater variation in traits and with greater spatial replication." I believe "data collection in species richness" implies a different meaning.

Finally, and this is a bit of a pedantic point... the definition of the heatwave period start and end is still not very clear. I understand that the start and end of the heat wave period are taken when there are more than 5 sites with heatwave extend (not heatwave days- this is still confusing the caption) but how are the heatwave extent periods for each site defined? You are connecting together over time the first and last heat waves and in some sites there was at least a gap of a month or more in between with no heatwave. I have no problem with the period used, but according to your own definition there are actually four separate heatwaves!

Reviewer #2 (Remarks to the Author):

I thank for authors for addressing my concerns and even re-running some of their analyses to demonstrate the robustness of their conclusions.

I have, however, still have a mild concern about how skillful the network really is in representing drought and heat load for the presented species. This concern is partly rooted in the distribution of measurement sites as listed in table S1, and partly rooted in the use of relative metrics for drought and heat load.

As an example, the majority of beech sites is located above 800 m a.s.l., which in my understanding can lead to the relatively optimistic response pattern for beech in Fig. 2. Many strongly affected beech stands in central and southern Germany are distributed along lower altitudes (own observations and e.g., Obladen et al. 2021), i.e., an altitudinal range that is not covered well by the network presented in this study. From dendroecological studies it is known that the detrimental effect of heat waves and droughts on tree growth in the low altitudes can be reversed towards a positive effect in higher altitudes (e.g. Dittmar et al. 2003) - can the authors exclude such a potential mis-attribution, e.g. that the adverse conditions by heat and drought in the low lands are actually translated into more favorable growing conditions at higher altitudes? In my understanding, the relative definition of heat and drought used in the study would not exclude such a misinterpretation.

Other than that, I am very convinced by the revision. Especially the rebuttal of my concern about a potential influence of xylem phenology on TWD dynamics is convincing, and I thank the authors for their well-argued clarification.

Reference:

Dittmar, C., Zech, W., & Elling, W. (2003). Growth variations of Common beech (*Fagus sylvatica* L.) under different climatic and environmental conditions in Europe—A dendroecological study. *Forest Ecology and Management*, 173(1–3), 63–78.

Obladen, N., Dechering, P., Skiadaresis, G., Tegel, W., Keßler, J., Höllerl, S., Kaps, S., Hertel, M., Dulamsuren, C., Seifert, T., Hirsch, M., & Seim, A. (2021). Tree mortality of European beech and Norway spruce induced by 2018-2019 hot droughts in central Germany. *Agricultural and Forest Meteorology*, 307, 108482. <https://doi.org/10.1016/j.agrformet.2021.108482>

Response to reviewers letter | Manuscript "The 2018 European heatwave led to stem dehydration but not to consistent growth reductions in forests" (NCOMMS-21-21559A)

Reviewer #1

The authors responded exceptionally well to my comments and I am happy to recommend publication of this manuscript.

Perhaps there are three small points that could still be considered.

Firstly, I find the fig. R4 and the formula for REW highly illustrative and wonder if this could be included in the ms methods (both the equation and explanation of the symbols, and the figure). Of course the reader can also look up the cited references but it would make it easier for the reader if it is included in the text.

Response: The equation has been included in the Methods section, and Figure R4 has been included in the Supplementary information (as Figure S7).

Secondly, should the added text in line 343/344 not read " .. yet requires data for more species with greater variation in traits and with greater spatial replication." I believe "data collection in species richness" implies a different meaning.

Response: "Species richness" has been omitted, and the sentence has been reworded accordingly.

Finally, and this is a bit of a pedantic point... the definition of the heatwave period start and end is still not very clear. I understand that the start and end of the heat wave period are taken when there are more than 5 sites with heatwave extend (not heatwave days- this is still confusing the caption) but how are the heatwave extent periods for each site defined? You are connecting together over time the first and last heat waves and in some sites there was at least a gap of a month or more in between with no heatwave. I have no problem with the period used, but according to your own definition there are actually four separate heatwaves!

Response: The explanation of the heatwave period (HW) was indeed not 100% clear. The explanation has been clarified. In the caption of Figure S1, we now make the distinction between site-specific and overall HW extent for analyses and further clarify that the overall HW extent connects the first and last DOY when more than five sites overlap in site-specific heatwave days (not site-specific heatwave extent):

"Figure S1 | Heatwave extent in 2018 for all sites included within the network. a-b, Frequency distribution and site-specific distribution of the heatwave days and extent and days. A heatwave is defined as five consecutive days with a maximum daily temperature higher than the 90th percentile of the control period (1951-2000 E-OBS data). The selected overall heatwave extent for analyses is indicated with dotted lines (from DOY 208 until 264). This overall extent shows temporally covers the first and last day of year (DOY) when more than five sites overlap in heatwave days to define the start and the end of the overall heatwave period, respectively."

I thank for authors for addressing my concerns and even re-running some of their analyses to demonstrate the robustness of their conclusions.

I have, however, still have a mild concern about how skillful the network really is in representing drought and heat load for the presented species. This concern is partly rooted in the distribution of measurement sites as listed in table S1, and partly rooted in the use of relative metrics for drought and heat load.

As an example, the majority of beech sites is located above 800 m a.s.l., which in my understanding can lead to the relatively optimistic response pattern for beech in Fig. 2. Many strongly affected beech stands in central and southern Germany are distributed along lower altitudes (own observations and e.g., Obladen et al. 2021), i.e., an altitudinal range that is not covered well by the network presented in this study. From dendroecological studies it is known that the detrimental effect of heat waves and droughts on tree growth in the low altitudes can be reversed towards a positive effect in higher altitudes (e.g. Dittmar et al. 2003) - can the authors exclude such a potential mis-attribution, e.g. that the adverse conditions by heat and drought in the low lands are actually translated into more favorable growing conditions at higher altitudes? In my understanding, the relative definition of heat and drought used in the study would not exclude such a misinterpretation.

Response: We perfectly understand this concern. Despite being the first continental-scale dendrometer dataset for large-scale analysis, we acknowledge the limited spatial resolution and species-specific replication of the compiled dataset. Of course, these limitations affect the output of the statistical analyses, and the stem growth response to environmental drivers will be site-dependent and specific-dependant. Actually, we believe the example outlined above for beech trees could be similarly argued for each sampled species within the dataset, as potential sampling bias is inherently associated with large-scale analysis, like the one presented here.

For this reason, aiming at delivering a message within the context of the collected dataset, we repeatedly mention throughout the manuscript that:

- "our results were dependent on the site selection for analysis",
- "limitations to stem-girth increment are therefore highly dependent on site-specific conditions", and
- "a variable response among sites and species cancelled each other out. Illustratively, regional studies have reported negative^{28,29}, neutral^{30,31}, and positive³² responses of annual stem growth to HW2018".

Therefore, we do not think additional clarifications are required in this line. In any case, our tree-level observations agree with ecosystem-level measurements of carbon exchange across Europe, as we discuss at the beginning of the Discussion section (i.e., spring growth stimulation and summer reduction compensated each other, resulting in nearly average ecosystem carbon uptake over the year, which implies a neutral growth response to the 2018 heatwave at the continental scale).

Other than that, I am very convinced by the revision. Especially the rebuttal of my concern about a potential influence of xylem phenology on TWD dynamics is convincing, and I thank the authors for their well-argued clarification.

Response: Thank you for this appreciation.

Reference supplied by Reviewer #2:

Dittmar, C., Zech, W., & Elling, W. (2003). Growth variations of Common beech (*Fagus sylvatica* L.) under different climatic and environmental conditions in Europe—A dendroecological study. *Forest Ecology and Management*, 173(1–3), 63–78.

Obladen, N., Dechering, P., Skiadaresis, G., Tegel, W., Keßler, J., Höllerl, S., Kaps, S., Hertel, M., Dulamsuren, C., Seifert, T., Hirsch, M., & Seim, A. (2021). Tree mortality of European beech and Norway spruce induced by 2018-2019 hot droughts in central Germany. *Agricultural and Forest Meteorology*, 307, 108482. <https://doi.org/10.1016/j.agrformet.2021.108482>